# Circular Economy and Financial Aspects: A Systematic Review of the Literature

**Beatriz de Souza Mello Gonçalves, Flávio Leonel de Carvalho \***  **and Paula de Camargo Fiorini** 

Department of Administration, Federal University of São Carlos (UFSCar), Sorocaba 18052-780, Brazil; beatrizmello.g@gmail.com (B.d.S.M.G.); paula.fiorini@ufscar.br (P.d.C.F.)
* Correspondence: flavio@ufscar.br

**Abstract:** The objective of this article is to analyze the pre-existing studies that investigate the link between the circular economy and financial aspects in order to understand the evolution of the circular economy literature and its relationship with finance. In addition, it proposes an investigation of empirical evidence of economic-financial gains resulting from the adoption of circular production practices. The methodology used to achieve this goal was a systematic review of the literature and bibliometric analysis. Thus, it was possible to conclude that the barriers faced by companies adopting the circular economy in relation to financial performance are defined by (i) the size of the business and the initial investment cost, (ii) difficulties for micro and small companies, (iii) to a more complex structuring of the business, and (iv) greater exposure to risk, as the circular economy is a new concept and is and not as representative as a linear standard system. The results show that few studies investigate corporate gains from circular production, which is, therefore, an important topic for future research and the major contribution of this paper.

**Keywords:** circular economy; finance; accounting; organization; financial performance; sustainability

## 1. Introduction

The current economic system is mainly based on a linear logic for obtaining capital: extraction of natural resources, production, consumption, and, finally, disposal [1,2]. As long as we live in an economy based on linear production and consumption processes, the destruction of ecosystems, the loss of biodiversity, and consequences such as climate change and water and air pollution will continue causing irreversible changes that render it difficult to sustain life on Earth [3].

An alternative to this model is a system that guarantees a continuous material flow, i.e., a closed loop. In this logic, the concept of circular economy comes into play, as it establishes a connection between the use of resources and waste disposal, thus converting the linear system into a circular system [4]. The circular economy can be defined as an economic system based on business models that replace the concept of linear logic through alternatives such as recycling, reuse, use of renewable energy, and product design, operating from the micro sphere to a macro perspective (eco-industrial parks, cities, governments), thus creating a balance between the environment and economic prosperity [5].

As the concept of a circular economy does not have a single definition—it is based on a collection of biases from different areas of study, such as environmental engineering, business, environmental sciences, among others [6,7]—it is still much debated and can be implemented in many ways. As well as the concept, the origin of the term is also studied and cannot be attributed to just one author or date. Although the momentum acquired in the 1970s can be registered, it has its own dynamics in different spheres [2]. The concept of a circular economy was introduced by Pearce and Turner [8], and the authors claim there is a strong interdependence between the economy and the environment when looking at their relationship through the first law of thermodynamics, which states that energy and natural

resources cannot be destroyed, only transformed. A simple and linear economic system does not consider this fact as it is constantly generating waste that does not decompose naturally and cannot be transformed into environmentally friendly products [8]. Using an opposite strategy to the current system, CE strives to face the challenge of resource scarcity and waste disposal in a win-win approach [9].

A circular economy has been a current practice in several countries. It has been implemented by nations such as China, Germany, Japan, and members of the European Union (EU), through laws that guarantee good waste management, recycling targets, and directives against waste [10–13]. In addition to the benefits to society as a whole, the practice of the circular economy may also result in economic gains to the companies, thus having positive social, environmental, and financial effects [14,15]. Adopting sustainable practices is an attractive action; however, many studies report that strategies to implement a circular economy are still in their infancy [16–18]. In addition, few studies clarify how to develop a circular business model, making it difficult to replicate it in other companies [19]. Likewise, the relationship between the circular economy and finance has not yet been fully studied [20], demonstrating the pressing need to analyze the impact of circular economy practices on the companies' economic and financial performance indicators, such as profitability, market value, capital cost, return on investment, production cost, etc. [21,22]. However, despite the growing importance of the circular economy in academic research— the number of studies on the subject had a significant increase in the last decade [18,21], little has been investigated regarding its impact on the performance of companies [20,23], showing a gap in the literature. In line with this argument, Camón Luis and Celma [24] found in a literature review that there is still a tendency to focus on the circular economy mostly from the environmental dimension, neglecting the financial and economic aspects. Additionally, Goyal et al. [21] revealed a lack of comprehensive research on CE metrics to provide an objective review of CE assessment frameworks and performance. Thus, once this perception of the financial benefits of a circular economy is evaluated, its attractiveness can be used as an incentive for its application in the business sphere.

Therefore, considering the importance of the theme from an economic, social, and financial point of view and the need to extend this area of research, the present study aims to analyze pre-existing studies on the circular economy and its relationship with finance. For that, a systematic literature review was conducted, using bibliometric and content analyses for the works assessment. Although there exist some bibliometric studies about the circular economy [14,21,24], they have approached different perspectives of the topic, such as mapping key research streams [21,24], and the role of the environment [14]. Furthermore, recent review works have attempted to explore the relationship of CE and performance but considering the digitalization context [15], and the development of a framework for circularity degree [22]. In this sense, this research contributes and expands the extant literature by disclosing findings of financial aspects related to CE, evidence of economic-financial gains arising from circular production practices, and financial barriers to the circular economy in companies. Thus, it brings managerial and theoretical contributions.

The remainder of this paper is organized as follows. This first section introduces the context and objective of the study, positing its importance in the literature. In the next section, the theoretical background of the research is presented. Section 3 displays the methodological steps taken to conduct the systematic literature review. Section 4 depicts the analysis and discussion of the results. Finally, Section 5 presents the conclusions of the study.

## 2. Theoretical Foundation

Some ecological economists are frequently cited in the debate about the origin of the term "circular economy" due to their strong influence on the construction of the concept over time. Boulding [25] points out that humans need to find their place in a cyclic ecological system, even if they are capable of continuously reproducing material, thus proposing a closed-loop system. Pearce and Turner [8] base their thoughts on Boulding [25] when

developing their idea of a cyclical production and consumption system, citing the term "circular economy" when they explain that the term points to the environmental limits of extracting natural resources.

The concept of a circular economy is composed of different fragments, and it is possible to say that it relates to and/or derives from influential ecological concepts, such as cradle to cradle, industrial ecology, industrial symbiosis, sustainable supply chain management, performance-based economy, and blue economy [26,27]. As such, it can contain different interpretations. For Yuan, Bi, and Moriguich [13], the three pillars of the circular economy are industrial ecology, clean production, and ecological modernization. The origin of the concept itself comes from the collaboration of several authors and has a range of meanings. However, it is correct to state that all associations to the term have the understanding of a closed-loop system [7]. Another practical concept that relates sustainability and cooperation through the circular economy, renewable energy, and quality of life is the slow city concept. It's becoming an important matter mainly in the European Union. The slow cities concept relates to the circular economy and aims to increase better living conditions in cities [28].

The circular economy is characterized as being regenerative, intensifying natural capital, optimizing stocks and the production of resources, and showing potential for innovation, job generation, and economic growth [2].

The circular economy can be promoted through government subsidies, effective legislation, economic incentives, and development and research [29]. Governments are important agents in this process, as they can prevent irregularities and unsustainable practices in industries through taxes and taxation, and offer subsidies to ecologically correct products [30]. Through effective legislation, products can be designed for recycling, thus resulting in a longer lifespan, and benefiting society and the environment [31]. The European Union is a reference in adopting sustainable directives. The Netherlands, for example, has promoted subsidies for product design and innovation, Belgium has adopted subsidies to encourage packaging recycling, and Austria has used subsidies to reduce pollution and waste [10], and Poland also has legal regulations to adopt a circular economy, although it faces a lack of organizational and financial solutions [32].

Still on the international scenario, Germany can be mentioned as an example of circular economy practices given that it enacted, in 1996, the Closed Substance Cycle Law and the Waste Management Act (Closed Substance Cycle and Waste Management Act) in order to preserve natural resources and ensure environmentally compatible waste disposal [12]. The Japanese government can also be cited for encouraging sustainable production and consumption practices, as they have created laws from 1995 to achieve recycling goals in a national plan to promote a more sustainable society [11]. Among these laws is the Basic Law for Promoting the Creation of a Recycling-Oriented Society, which aims to prevent waste generation and guarantee proper waste disposal. Since then, the 3R's (reduction, reuse, and recycling) have been applied to different materials [11,33]. In China, the concept of a circular economy was identified as a national political objective in 2002 [13]. Given the huge neglect of the environment and the serious environmental problems faced by the country, the concept of a closed-loop was seen by the government as a strategic alternative [13,34], and China currently concentrates a large number of studies on the subject. The European Union can also be mentioned for adopting directives against waste production from the 2000s onwards, such as recycling, prevention and reduction of negative effects in landfills and the correct disposal of batteries, among others [10]. Based on these cases, the implementation of national circular economy practices appears to be recent. Such practice also depends on a massive adherence by society and change in consumer awareness, requiring integration between national leaders, companies, and consumers in order to create a path to the future of humanity [29].

In the organizational and financial dimension, Aranda-Usón et al. [20] aim to describe the influence of financial resources in circular economy practices in organizations. To that end, a survey was carried out in partnership with the Spanish Ministry of the Economy. The study identified some financial barriers to the transition from a linear to a circular

economy. Among them, the authors cite the lack of assistance from public institutions; insufficient investments; and the size of the business, a crucial factor for the viability of a circular system. Being beneficial, the link between circular economy and finance can have a beneficial effect on the course of financial viability of companies in order to generate a great impact on society and the environment. Based on this, the present paper aims to synthesize preexisting studies on the circular economy and its relationship with finance or accounting.

Studies that analyze the impact of adopting circular economy practices on the financial performance of companies are still incipient [20,23]. Measuring the financial benefits of the circular economy and the financial profitability of companies that adhere to it are some of the barriers identified with regard to the adoption of the closed system by organizations [35] The current state of circular business models conveys the idea of a risky venture, in which the main issue is the added value of implementing the system. However, in the long term, a shortage in the supply of natural resources is expected to happen, forcing large companies to adapt to a circular system [36].

Corporate sustainability, a long-debated topic, has received more in-depth attention than the circular economy. Sustainable conducts are expected to add value to brands and increase market competitiveness. However, studies that compared the sustainable indexes of stock exchanges in some countries and analyzed the relationships between financial and environmental performance showed that there is no significant financial advantage between companies that adopt socially responsible attitudes and the ones that do not [37–39]. So, since profit maximization is one of the companies' objectives to create value and attract investors, would it be feasible to exercise social responsibility and reduce returns? [40]. Other works, however, highlight positive aspects between the association of socially responsible investments, corporate sustainability and finance, stating that this factor does not harm investors' interests. Instead, the authors state that it provides long-term financial leverage even if it is more sensitive to market variations and has a tendency for financial risks [41,42]. It's important to mention that socially responsible investments are still under investigation, as studies show conflicting results. The results of these surveys depend on the location of the object studied, as the performance of different companies depends on macroeconomic aspects and also on the profile of the investor [41] and/or consumer.

The works that analyze corporate sustainability and finance present different conclusions. Some show that it is not possible to find statistical or financial advantages on the part of companies with socially responsible attitudes [37–39], while others indicate a positive relationship between companies that take this stance [41,42]. Therefore, the need for research that seeks to investigate the relationship between a circular economy and the financial viability of organizations is evident. Thus, the objective of this work is to analyze and group preexisting studies in the area through a bibliometric review and to contribute to the literature on circular economy and finance or accounting.

## 3. Materials and Methods

This research included a systematic literature review (SRL), which consolidates the results of different studies on a subject with the aim of achieving greater understanding and reaching a level of conceptual or theoretical development [43]. For performing the SRL, bibliometric and content analyses were used. Thomé et al. [44] explain that a qualitative SLR is often combined with quantitative methods, such as the bibliometric analysis to identify and visualize emerging themes. The approach and steps taken followed the instructions for SRL from the studies of Thome et al. [44], and Tranfield et al. [45]. First, the review scope and purpose were clearly defined, as well the database, keyworks, and criteria for the searches were chosen. Second, the search strategies were put in practice. This is, the search for articles was conducted in the database and the predefined filters and criteria were applied to the sample. In the sequence, all the abstracts were read to define the final sample. The third step involved performing a bibliometric analysis with the purpose of identifying the evolution of research topic, discover the most relevant articles and authors,

countries and institutions to understand the stage of development of the theme [46]. Next, in the fourth phase, a content analysis was carried out to deepen the knowledge on the topic by synthesizing the main contributions of the studies in the analyzed sample. For that, a data gathering template was created to ease the extraction of the information [44]. Finally, the data retrieved from the bibliometric and content analyzes were interpreted and discussed. Figure 1 summarized the phases of the SRL, and the details of each phase are explained in the following sections.

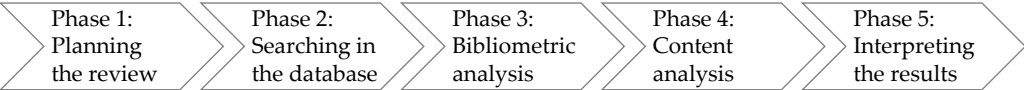

**Figure 1.** General phases of the SRL. Based on Thome et al. [44].

### 3.1. Planning the Systematic Literature Review

In this phase, the scope of the review was defined. That is, the objective was to analyze the existing literature relating the circular economy and financial aspects. For finding these studies, the database chosen was Scopus (Elsevier). The Scopus platform was selected because of its vast collection with more than 73 million records in 24 thousand journals [47]. Besides that, Scopus is one of the most used databases in the academic world [48–50].

Regarding the research keywords, the following search terms and Boolean operators were defined: "Circular Economy", as the first search field, and "Finance" OR "Accounting" OR "Financial Performance" OR "Economic Performance" OR "Green Finance" as the second search field. It is important to point out that the research including the two search fields was essential to cover the topic more broadly and carry out more thorough research.

### 3.2. Searching in the Database

Data were collected on 30 March 2021, resulting in an initial sample of 308 documents on the Scopus platform. After applying the filter of type of document, articles published in congresses and conferences, book chapters, books and errata were excluded, resulting in 246 documents. In this first stage, documents such as articles, reviews and editorials were kept. Subsequently, only papers from journals were considered, returning 244 results. Finally, records were limited to the English language, excluding Russian and Chinese, showing 233 scientific publications. Table 1 presents the applied search criteria and the number of documents that remained in the process of the conducted analysis.

**Table 1.** Search criteria in Scopus platform.

| Search Criterias | Entry | Results |
|---|---|---|
| Search field | Article title, Abstract, Keywords | - |
| Keywords and boolean terms | "Circular Economy" AND "Finance" OR "Accounting" OR "Financial performance" OR "Economic performance" OR "Green finance" | 308 |
| Filter 1 | Document type: Articles, Review, Editorial | 246 |
| Filter 2 | Source type: Journal | 244 |
| Filter 3 | Language: English | 233 |
| Total documents | | 233 |

The next step involved the reading of the abstracts of the 233 resulting scientific publications. Only the articles approaching the scope of the study were selected. As a result, a total of 69 articles were considered relevant to the research. The number of papers considered in the dataset is suitable, considering similar studies that applied systematic review and bibliometric analysis, such as Thomas and Gupta [51] with a sample of 59 documents and Ferreira et al. [52] with 53 articles.

### 3.3. Bibliometric Analysis

The bibliometric analysis consists of "the application of mathematics and statistical methods to books and other media of communication" [53] (p. 349). In summary, it is a quantitative analysis of scientific publications, through which academic literature in relation to a given topic is mapped [54]. The bibliometric method analyses the literature observing the academic performance and mapping the relevant publications, authors, institutions and terms in a way that they can be interpreted and inferences can be made [55,56]. Besides, it applies rigor to the analysis and increases the objectivity of the research. Thus, the present work used the bibliometric analysis to synthesize the preexisting studies, map the scientific research field, and understand the stage of development of the theme.

The software used to assist data visualization and bibliometric analysis was the VOSviwer®, as it offers a detailed reading of bibliometric maps, in addition to supporting maps with more than 10 thousand items [57]. Furthermore, Pan et al. [58] found that VOSviewer is more frequently used and diffused than CiteSpace or HistCite in bibliometric studies [24,59]. Mendeley software was applied to read and organize the works analyzed, and Microsoft Excel to generate graphs outside the database.

Figure 2 presents the bibliometric analysis steps. At the first step, the data of the 69 publications were retrieved from the Scopus platform in the Excel format (.csv). In the second stage, the .csv file was incorporated into the VOSviewer software for running the analyses. For ensuring the reliability of the data, a data cleaning was conducted, and some references were formatted so that there would be no duplication of authors' names later. At this step, a VOSviewer Thesaurus (.txt) file was produced and used to format the data and to avoid any duplicate results during network analysis. The third stage comprised the elaboration and analysis of networks, aiming to develop the bibliometric review. In the last stage, works with greater impact on the network map were read in full.

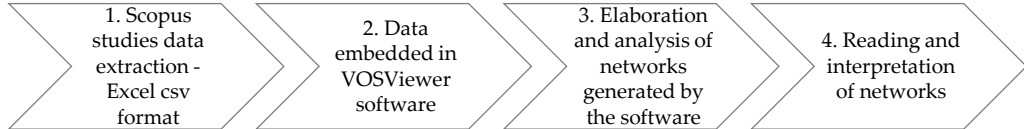

**Figure 2.** Bibliometric analysis steps.

The present study focused on three types of networks analyses. First, a keyword network was created to determine the most cited words in the publications. As a result, it was possible to identify the trends and the scope of the theme, showing how many times the different keywords were mentioned in the sample [55]. Then, a cocitation network was elaborated to obtain the theoretical pillars of the subject. The cocitation network identifies the documents that are cited together in the sample's works, that is, in case a third work references two others. [60]. The greater the apparent node in the cocitation network, the more a publication is identified as relevant to the topic, and the proximity of the nodes determines the degree of similarity between them. Subsequently, the citation network was produced. This includes the most influential articles, as it provides a view of most cited works [61]. The larger the apparent node on the map, the more the publication was cited. This section may be divided by subheadings. It should provide a concise and precise description of the experimental results, their interpretation, as well as the experimental conclusions that can be drawn.

### 3.4. Content Analysis

After the network analysis in VOSviewer, the full articles were downloaded and incorporated into the Mendeley software, which helped with the organization and their full reading for the content analysis phase.

For Seuring and Gold [62], a content analysis "represents an effective tool for analysing a sample of research documents in a systematic and rule-governed way". While a content

analysis can be performed using a quantitative or qualitative approach, this study followed the qualitative one to provide insights from summarizing the main contributions of the articles analyzed. Following the steps of Mayring [63], the process for a qualitative content analysis involved: (1) delimitating the sample of works; (2) assessing the formal characteristics of the articles, which consists in a descriptive analysis; (3) selecting a classification and categories for data extraction; and (4) analyzing the sample using the defined scheme. Therefore, the 69 articles were read in full and data extraction was performed using a coding scheme, which approached descriptive and content-related fields. The coding scheme encompassed six main categories, namely: author's name, title of the article, purpose of the study, research method applied, type of financial aspect approached in the study, and main contributions to circular economy and financial aspect. Through the analysis of the articles, it was developed and is detailed in Appendix A.

### 3.5. Interpreting and Presenting the Results

At the final phase, the results of both quantitative and qualitative analyses were interpreted and discussed. The next section reveals the findings of this study.

## 4. Analysis and Discussion of Results

### 4.1. Temporal Analysis

Figure 3 presents a graph with the evolution of the annual publications of works related to the circular economy, finance, accounting, financial performance, economic performance and green finance after filtering. The graph shows a growing trend in publications. In 2020, a peak with 27 publications was reached. It also shows that the first publication on the topic was in 2012, pointing out that it is a recent topic which has grown over the years. It notable that the number of studies on the topic had a fundamental increase in the last decade due to the importance of the subject [18].

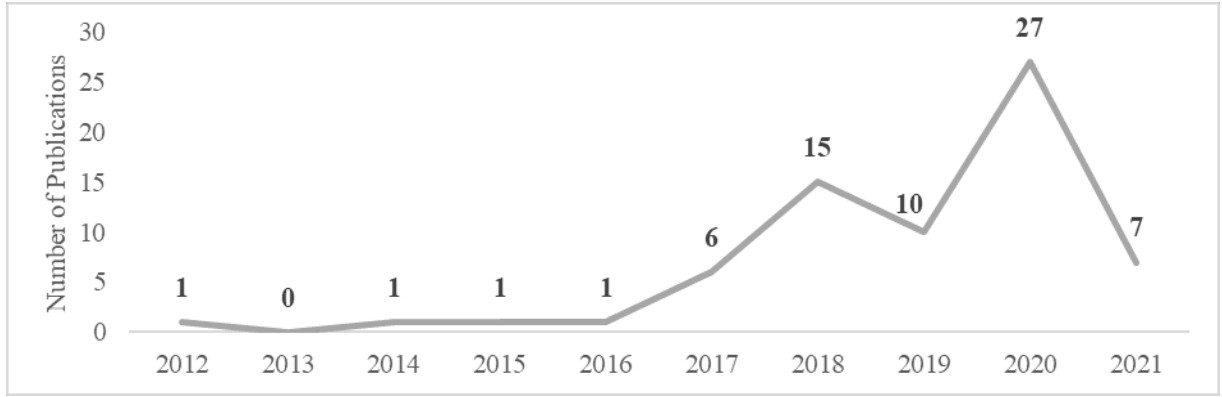

**Figure 3.** To March 2021.

### 4.2. Analysis of Documents by Publication Area

Fifteen areas of knowledge with scientific papers on the circular economy and finance or accounting were verified. Of these, environmental sciences alone was the focus of 25% of the publications. The percentages were: environmental sciences (25%), business, management and accounting (18%), energy (14%), business engineering (13%) and social sciences (10%), which comprise approximately 81% of the total of works produced and make up the areas with the greatest predominance of research on the subject. Table 2 shows the values in percentage for better visualization, as it is common for two or more areas to act together in research.

**Table 2.** Distribution of publications by area.

| Area | Number of Publications | % |
|---|---|---|
| Environmental Sciences | 45 | 25% |
| Business, Management and Accounting | 32 | 18% |
| Energy | 26 | 14% |
| Engineering | 24 | 13% |
| Social Sciences | 19 | 10% |
| Economy, Econometrics and Finance | 15 | 8% |
| Computer science | 5 | 3% |
| Others | 15 | 8% |

*4.3. Analysis of Leading Journals, Countries and Institutions*

Table 3 shows that the five journals with the highest number of publications on the topic are the Journal of Cleaner Production (13 publications), Resources Conservation and Recycling (five publications), Sustainability Switzerland (five publications), Journal of Industrial Ecology (four publications), and Business Strategy and The Environment (three publications). It is important to emphasize that the research studies in these important scientific journals were produced between 2015 and March 2021.

**Table 3.** Publication distribution by periodicals.

| Journal | SCImago Journal Rank (SJR) 2020 | H-Index | Number of Publications |
|---|---|---|---|
| Journal of Cleaner Production | 1.94 | 200 | 13 |
| Resources Conservation and Recycling | 2.47 | 130 | 5 |
| Sustainability Switzerland | 0.61 | 85 | 5 |
| Journal of Industrial Ecology | 2.38 | 102 | 4 |
| Business Strategy and the Environment | 2.12 | 105 | 3 |
| Amfiteatru Economic | 0.34 | 20 | 2 |
| International Journal of Life Cycle Assessment | 1.09 | 105 | 2 |
| Others | - | - | 35 |

Table 3 also includes SCImago Journal Rank (SJR) and H-Index, which are indicators that assess scientific journals. SJR indicates the prestige of a journal's impact, and H-index represents its articles number [64]. The indicator values were based on Scopus data as of April 2021.

Regarding the distribution of publications by countries, Table 4 shows that China ranks first, with 11 documents. This can be explained by the strategy of the Chinese government to promote sustainability and the circular economy, given the pollution problems faced by the country [13]. Next are Italy, the United Kingdom, the United States, and Spain, with 10, 8, 8, and 7 scientific publications, respectively.

**Table 4.** Publication distribution by country.

| Country | Number of Publications |
|---|---|
| China | 11 |
| Italy | 10 |
| United Kingdom | 8 |
| United States | 8 |
| Spain | 7 |
| Denmark | 5 |
| Germany | 4 |
| Others | 180 |

In Table 4, it is notable that China concentrates the largest number of publications, which can be explained by the serious environmental problems faced by the country, as the

circular economy is an alternative for the Chinese government in reestablishing the socio environmental area [13,34]. Despite the United States and China, it can be observed that all countries listed are found in Europe, and some belong to the European Union, which can be explained by the European Union being referenced in adopting sustainable directives [10].

The distribution of publications by educational institutions is quite wide. It is important to point out that some works are carried out jointly by different research institutions, changing the original total of 69 publications. The same is illustrated in Table 5.

**Table 5.** Distribution of publications by research institution.

| Institutions | Number of Publications |
|---|---|
| Universidad de Zaragoza | 7 |
| Centro de Investigación de Recursos y Consumos Energéticos | 6 |
| Universidad de La Rioja | 3 |
| Central South University | 3 |
| Worcester Polytechnic | 2 |
| Geoponiko Panepistimion Athinon | 2 |
| Aalbourg University | 2 |
| Others | 146 |

Table 5 shows that the institution with the highest number of publications is the Universidad de Zaragoza (7 publications), followed by the Centro de Investigación de Recursos y Consumos Energéticos (6 publications), Universidad de La Rioja (3 publications), Central South University (3 publications) and Worcester Polytechnic (2 publications).

*4.4. Co-Words Analysis (Keywords)*

The keyword network presents the information provided by the authors of the publications. Figure 4 demonstrates the network of keywords generated from the sample extracted from Scopus and includes terms with at least five occurrences, that is, the number of publications in which the words occur together. The term "article" was excluded from the network, as it was not related to the topic. There are three groups: red, green and blue. The most popular words are "circular economy", "sustainable development", "recycling", "sustainability" and "economics".

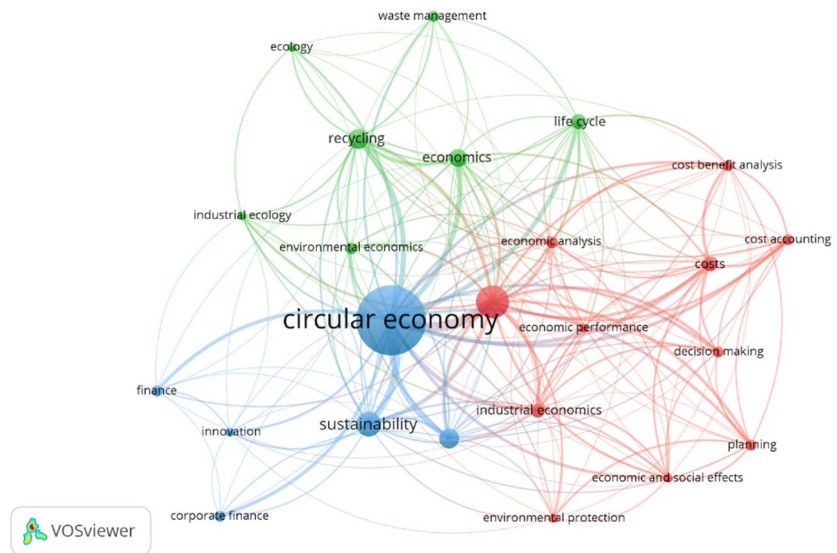

**Figure 4.** Keyword Network.

Regarding the groupings, the green cluster groups terms such as "economics", "waste management", "life cycle analysis" and "recycling". The most relevant works in this group

that mention the circular economy is the article by Haas et al. [65], with 330 citations. The aim of the article is the analysis and discussion of the global and European Union circularity of material flows in 2005, and it mentions energy derived from fossil fuels and metals, among others. The result shows that the European Union has a slightly higher index than the rest of the world in terms of recycling and circularity of materials. The study asserts the need for "ecodesign" (sustainable design) resulting in renewable energy, recycling and product design for the circular economy to advance. The second most cited article in this group is that by Pauliuk S. [66], which states that faulty indicators can be a risk for circular companies. This group has publications that analyze the flow of materials used in different segments and address future environmental consequences. Most studies in this group present sustainability strategies to avoid the depletion of finite natural resources [65–69].

The blue group contains words such as "circular economy", "finance" and "innovation". The article by Scarpellini et al. [70], cited 24 times, relates financial performance with the implementation of the circular economy in firms, while Dobrota, Dobrota and Dobrescu [71] focus on finance in relation to eco-innovations or new technologies. These studies investigate the preexisting theory, look for alternatives and analyze the barriers to the implementation of the circular economy in terms of financial feasibility, lifespan of materials and organizational processes.

Specifically in relation to finance, these works identify initial investment costs as being potential obstacles to circular economy initiatives, highlighting the importance of research and guidance on the financial feasibility of projects in this area, given that the structuring of circular business models demands more time [70,72,73]. They also mention the importance of public financial incentives (subsidies), which ensure both the reduction of exposure to risk and the profitability of certain projects. The works finish by highlighting the need for further research, as this is a new and important topic for the development of the circular economy.

In the red group, Svensson and Funck [74], with 17 citations, assess the environmental and economic feasibility of adopting circular economy practices, since both waste recovery and product management require investments, implying the need for new action plans, accounting and investment evaluations. As the circular economy does not have a single meaning and can be approached in different ways by organizations, Svensson and Funck [74] investigate the relationship between product life cycle, management control and business models that adapted to the circular economic system. For this, the authors use three case studies and interviews as their research method. The results show that, although there are several ways in which circular economy can be used in an organization, it is very important for management, organizational culture, planning and strategic objectives to be aligned with circular principles. On the other hand, the authors emphasize that the level of detail with regards to the control of production and investment costs are higher when it comes to expanding the product's lifespan. Therefore, as it is a challenge for the long-term planning of circular businesses, this is a topic that should be further investigated. Their study also points out a challenge for research in relation to financial accounting control, process costs and product lifespan, such as calculating the cost of recycled material, as resources follow infinite life cycles for different products.

It is important to emphasize that the word "accounting", an initial search term, does not occur in the generated network, although it is mentioned many times as a possibility of study by the sample articles [69,72,74–76]. Svensson and Funck [74] even mention accounting as a challenge for future research on the circular economy, which demonstrates the need for studies on the relationship between circular economy and accounting. "Finance" occurs together with "circular economy" only six times in the sample, which demonstrates the need for further research relating the two terms.

To sum up, the theme of circular economy and financial performance covers different areas and dimensions, such as waste recovery, product management, cost evaluations and investments related to product life cycle, proper accounting of recyclable material flows and alternatives to linear production steps. All these topics must be investigated if the

circular economy is to be an option for organizations to replace the current system. The financial performance of circular business models is not limited to process costs, so financial viability is a success. It is necessary that in a previous stage, the product is analyzed in all its production stages and with regard to the different resources coming from recycling, reuse, and product design, among others.

*4.5. Cocitation Analysis*

The co-citation network (of the documents) demonstrates the co-occurrence of articles in the sample's reference list, that is, two publications are co-cited. Figure 5 demonstrates the cocitation network generated based on the sample. The parameters adopted were (i) references and (ii) articles cited at least three times. Two groups were observed: red and green. In addition, we decided to show only items that had a connection.

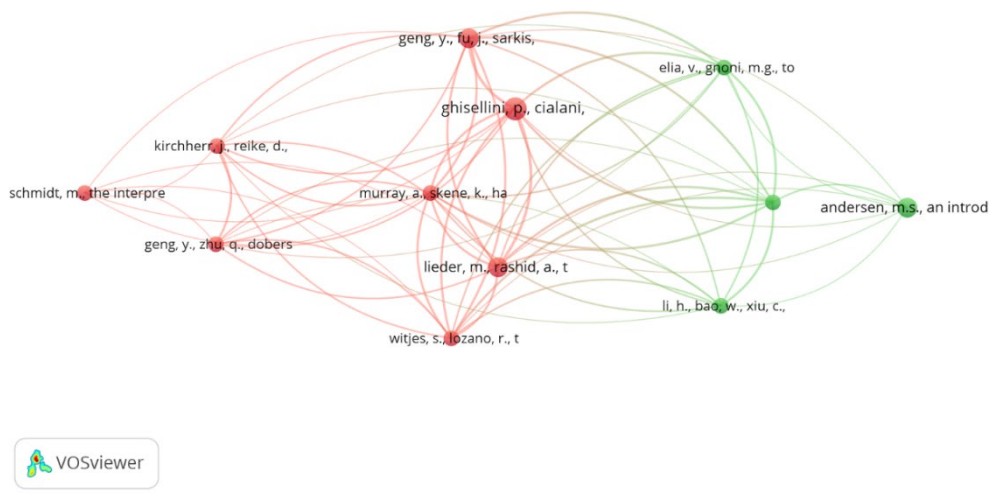

**Figure 5.** Cocitation Network.

The spare nodes indicate the theoretical pillars of the circular economy and its relationship to finance in general. The biggest node represents the study by Ghisellini, Cialani and Ulgiati [77]. Cited five times, as shown in the cocitation network, this work consists of a literary review of works from the last two decades, identifying the characteristics and perspectives of the circular economy, its origins, principles basics, advantages and disadvantages, modeling, and its implementation at different levels. The authors claim that the implementation of the circular economy is rooted in the ecological industry, a term that is mentioned several times in the text. The article also highlights the importance of taxation by the governments on non-renewable resources and incentives through financial subsidies that positively stimulate the circular economy and industrial symbiosis. In addition, the study presents several alternatives and examines the circular economy as a solution to businesses and the environment. The authors conclude by pointing the importance of creating an awareness among consumers, cities, legislation and countries, as it is crucial for the success of circular economy and the consequent preservation of the environment.

Industrial symbiosis is crucial for the ecological industry, as it constitutes a tool for the implementation of the circular economy [78]. The concept of ecological industry can be described as an industrial system that is not isolated, but connected to other industries (industrial symbiosis) and that applies product design in favor of the environment, preventing pollution and adopting circular models [79,80].

The study by Geng et al. [81], appearing with 4 citations in the cocitation network is a critical analysis of China's national circular economy indicator system. The study exposes the urgency of developing appropriate indicators to assess the development of different businesses so that a circular economy model can be implemented. Barriers to the effectiveness of the circular economy are also presented. The results indicate that the

existence of social, industrial, and material indicators in relation to clean production could be beneficial to the environment and to Chin, and, therefore, to society.

The authors mentioned in the paragraphs above belong to the red group, the largest group in the cocitation network, which is comprised of eight works. The majority of the articles in this group aim to elucidate the definitions and origins of the circular economy, relating different concepts such as ecological industry and industrial symbiosis [77]. The studies also raise possibilities for achieving sustainable development through the circular economy, pointing to it as an integration between economic activity and the "well-being" of the environment. The studies use different methods (theories based on data, interviews, the index method with a systematic approach, among others), but they all aim to explain the conceptualization of the term and possibilities for its development.

Andersen [1], with four citations in the generated network, is the author of the most cited article in the second group (the green group). It aims to highlight the fundamental principles of environmental economics and sustainability, leading to the fundamentals of the circular economy. With the same bias, the study by Elia [17] makes a literature review to present the state of the art of the circular economy and propose measures to monitor strategies related to its development. Another document from the green cluster is the study by Huysman et al. [82], which focuses on circular economy performance evaluation indicators aiming to guide decision-making, with a specific focus on the technical quality of plastic waste. The works presented in the green cluster were identified as the basis for understanding the circular economy and its principles, being of great relevance to the development of this study.

*4.6. Citation Analysis*

The citation network, or citation analysis, presents the most cited items in the sample. The greater the proximity between nodes, the more the works tend to be related. Figure 6 shows the citation network generated from the research relating circular economy, finance, accounting, financial performance, economic performance and green finance. The network was limited to articles with at least 3 citations, and that were connected to each other. It is important to note that the same figure is generated when choosing a minimum of 4 and 2 citations, demonstrating the incipience of studies on business finance and the circular economy. We chose to show only connected items.

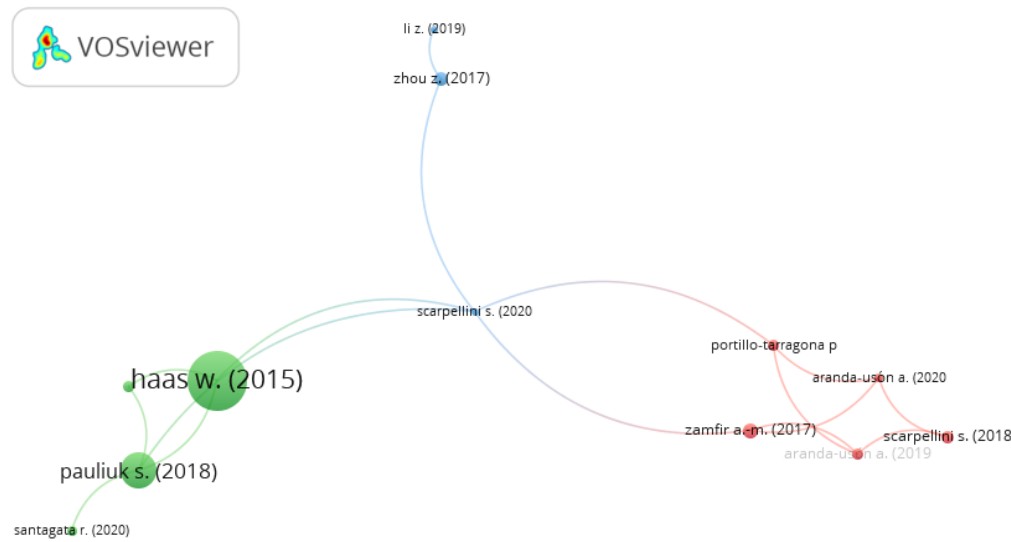

**Figure 6.** Citation Network.

The citation network above was created in order to better understand the relationship between the search terms and obtain the most influential publications for the study. The

cocitation network was then analyzed to obtain the theoretical pillars of the studied topic. The term "accounting" was not found.

Figure 6 presents 12 items and three groupings: green, blue and red. The biggest items, i.e., the most cited publications, are located in the green group. The study by Haas et al. [65], the most cited work in the sample (330 citations), explores the level of global circularity through a systemic perspective, using a definition of indicators that results in circularity in different nations and in the European Union. To do this, the main groups of materials and their due flows, from their extraction to their disposal, for the year 2005, were tracked. The study concludes that the use of fossil sources is a problem for the circularity of resources and highlights the importance of reducing barriers to recycling in order to extend product lifespans. Product design is also mentioned as a way to improve overall circularity.

Also in the green group, Pauliuk [66], the second most cited study in the citation network (141 citations), evaluates a technical standardization on the circular economy for companies. This technical standardization was prepared by the British Standards Institution, a global institution responsible for creating standards to improve work effectiveness in companies [83]. The study points out that the monitoring and strategy of circular economy were not met by the technical standardization evaluated. It also states that, with regard to the life cycle of materials, circular economy indicators must refer to the depletion of natural resources, stock growth and the useful life of the materials. The author also points out that inconsistent indicators on circularity may be a risk for companies that adopt the circular economy.

Figure 6 shows that the green, blue and red groups are mainly connected by Scarpellini S. [23], the blue grouping, which encourages the existence of a positive relationship between the circular scope of companies and their corporate social responsibility (CSR) and accountability. In addition, the author indicates that the relationship between the circular economy and the financial performance of companies, as well as the relationship between companies and circular economy practices, should be more thoroughly explored, as the existing studies on these themes are at an early stage [23]. To reach this conclusion, a study by Scarpellini [23], which is located in the center of the image and is the link between the three groups, aims to define the environmental capacity in a business since the concepts of circular economy are introduced in the event that the authors used as a research method the modeling of partial least squares structural equations.

In the center of Figure 6, in the blue group, the article by Zhou Z. [84], cited 27 times, uses the material flow cost accounting (MFCA) model as a research method, which is widely used for calculating industrial process costs. The authors examine how to modify this model by adapting it to a circular, biased system for the iron and steel industry. The work provides the necessary technical support for industries in the iron and steel sector when it comes to decision making, internal controls and cost accounting [84]. The third publication in the blue group is the one by Li Z. [85]. Although it only has 9 citations, the work relates to the others in its group as it also analyzes businesses. The article uses a case study to analyze the value stream related to the use of resources from the perspective of the circular economy, developing an extension of material flow cost accounting (MFCA) and modifying it by accounting for environmental damage, as well as economic benefits, and concludes that the analysis of the resource value stream can both reduce resource consumption and minimize environmental damage, reinforcing the option of sustainable development in industries [85].

The red cluster was identified as the most important for the object of study. The group brings together publications with a financial bias in corporate businesses, describing the relevance of financial resources to achieve corporate sustainability. The four publications that make up this group date from 2017 to 2020, demonstrating, once again, that the studies are recent and can be further explored.

Among the publications in the red group are the studies by Portillo-Tarragona et al. [86] and Scarpellini et al. [70], which aim to measure the capacity of financial resources applied to eco-innovation investments in companies. The works identified relationships between

managerial strategy and corporate social responsibility, as well as pointed out the importance of public financial incentives for companies that have eco-innovation processes to reduce their exposure to risk. The studies contributed by enhancing the knowledge about the impacts of resources and investments in eco-innovation projects. However, they present some limitations regarding the research method through which specific eco-innovation projects were evaluated. To tackle these limitations and obtain more satisfactory results, the authors propose a long-term research plan using a larger number of companies, including companies from other countries.

The article by Aranda-Usón et al. [20], also in the red group, aims to analyze the influence of financial resources applied to circular activities in companies. The study addresses the benefits of the circular economy and presents some financial barriers to its adoption, such as the lack of financial assistance from public institutions. It also states that the size of the business is a relevant factor in the transition to a circular model and that the smaller the business, the greater the difficulty to obtain investments, thus increasing risk. The article quantitatively analyzes a sample of companies with the intention of improving knowledge about resource management. It concludes that financial resources are still a barrier for adopting the circular economy, especially by organizations.

The studies by Aranda-Uson [87] and Zamfir [88], represented in the red group, relate business decision models with the adoption of circular economy practices. Zamfir [88] focuses on European micro and small companies and points out that the most important factor when it comes to adopting circular practices is the country where the companies operate.

*4.7. Systematization and Contributions of the Analyzed Studies*

Appendix A presents a summary, obtained through a systematic review of the literature, of the pre-existing articles that investigates the link between the circular economy and financial aspects. Based on that, the most relevant papers on this subject were analyzed. Thus, it was possible to conclude that the barriers faced by companies adopting the circular economy in relation to financial performance are defined by the size of the business and the initial investment cost; difficulties for micro and small companies, to a more complex structuring of the business and to greater exposure to risk, as the circular economy is a new and still not as representative as a linear standard system.

According to Aranda-Uson et al. [20] the size of the business is a crucial factor for the viability of a circular system. The smaller the business, the greater the difficulty to obtain investments will be. Another point is the cost of the circular production process that will be influenced by the size of the companies [89], perhaps because of the scale gains. Thus, one way to promote the adoption of circular practices by small and micro companies could be the implementation of public policy and tax incentives to stimulate this kind of initiative. In this way, based on previous literature, it is possible to obtain relevant research questions. First, what is the impact and importance of public policy on the economic viability of circular economic initiatives for micro and small companies? What is the importance of public financing lines for circular economic production practices in small and micro companies?

Additionally, the difficulties faced by a more complex structuring of the business that is required by the circular economy process could be mentioned, as could the difficulty of how to measure the impact of adopting circular economy practices on the financial performance of companies [20,23]. The current state of circular business models conveys the idea of a risky venture and the high-cost level of implementation, in which the main issue is the added value of implementing the system. However, in the long term, a shortage in the supply of natural resources is expected to happen, forcing large companies to adapt to a circular system [36]. Thus, the anticipation of this process could be a competitive advantage. From that, important questions need to be analyzed. Two points that needs to be addressed are how to incorporate future economic benefits from the circular production process on the business valuation and how to disclose these initiatives on accounting reports. The investments in the circular production process could be disclosed as intangible

assets, considering that benefits for the companies and society will be measurable only in long term. Furthermore, how does one recognize the negative externalities of not adopting circular production practices?

Finally, the link between the circular economy and financial aspects is related to the risk. Some studies highlighted the greater exposure to risk, just because the circular economy is a new system of production. However, the linear system has risks as well, and the environmental risk that it represents needs to be considered. From that, how does one weigh the risks associated with the circular economy with the environmental risks that the standard system represents? Is it possible to create some financial ratio or financial report that discloses and measures those aspects in a less subjective way?

Besides these main topics, a large number of articles confirm the lack of indicators to measure circular economy performance or that bring supportive findings to policymakers [68,87,90–93]. Other analyses identified other topics related to the circular economy but were not focused on the corporate finance matters, and included topics such as the life cycle of the materials [67], financial eco-innovation projects [86], green and smart cities [94], blockchain technology related to the circular economy [95], eco-efficiency accounting method and its applications [96], etc. The results presented in Appendix A show that few studies investigate corporate gains from circular production, which is, therefore, an important topic for future research.

## 5. Conclusions

This study aimed to analyze pre-existing studies on the circular economy and its relationship with finance, as well as to investigate the economic-financial gains, barriers to the companies' financial performance, and relevant aspects for the success of the circular economy in organizations by pointing out different guidelines for this type of economic practice. To that end, the study sought to analyze the most relevant articles in the literature in relation to the theoretical pillars, study trends and dimensions of the circular economy in different areas. The circular economy can be used in different ways and at different levels, in industrial processes, organizations, governments, and legislation. It does not have only one form of applicability. When implementing the circular economy in organizations, it is necessary that circular principles are aligned with the company's strategic objectives and organizational culture.

The literature presents a few papers mentioning the supply chain and production stages, given the vast potential that the circular economy has to adapt to stages such as reuse, recycling, ecodesign and product design. A tool that helps insert the circular economy into organizations is industrial symbiosis, which integrates production hubs and connects industrial systems from different companies so that there is sustainable exchange and incentives, and less pollution and environmental degradation.

To sum up, it is possible to conclude that the barriers faced by companies adopting the circular economy in relation to financial performance are defined by (i) the size of the business and the initial investment cost, (ii) difficulties for micro and small companies, (iii) a more complex structuring of the business, and (iv) greater exposure to risk, as the circular economy is new and still not as representative as a linear standard system. In addition, an important point that hinders the implementation of the circular economy is the lack of financial, organizational, and national indicators to assess the development of different circular businesses. It is important to point out that prior to financial success, the analysis of the product's costs in all its production phases needs to take into consideration resources from different sources (reuse, recycling, product design, etc.). Thus, there is a need for accounting control of process costs, since resources for different products can have different life cycles. Therefore, factors like financial incentives, subsidies for the projects, and the awareness of nations, companies and consumers are of great importance for the evolution of the circular economy.

Given that the theoretical environmental justification is extremely positive for the adoption of the circular economy and that, gradually, companies and large corporations

will need to comply with environmentally responsible measures, it is necessary to have financial indicators that can reflect the beneficial impact on value market shares. And as long as corporate sustainability does not go against the investors' interests, this will be the main lever for the adoption of sustainable management practices, which may be the economic incentive that is lacking and, consequently, a catalyst for the change from the linear to the circular economy. For this it is necessary to overcome the current challenges and barriers of costs, market value and process optimization.

Some limitations to the present study are the complexity and difficulty in detailing financial aspects. Given the methodology used and the incipience of studies on the financial performance of companies and the circular economy, only a few barriers need to be identified. No studies analyzing the direct impact of adopting circular economy practices on company performance (economic-financial performance) were found. Likewise, no works addressing the adoption of the circular economy and analyzing the way in which the results obtained are presented were detected. These points need to be investigated, as they can provide insight for future research.

Although important concepts that soften the subject have been synthesized, continuous research on the circular economy and its impact on the financial performance of firms is fundamental. These are responsible for progressively making the topic more recognized and for contemplating, in minute detail, the financial and especially the accounting relationship to the circular economy, which is seen little in the research sample. These are complex areas that demand operational strategies that are fundamental to the success of the business. Once related to the beneficial impact of the circular economy, they can potentially be a catalyst for its development and more widespread adoption.

**Author Contributions:** Conceptualization, B.d.S.M.G. and F.L.d.C.; methodology, B.d.S.M.G., F.L.d.C. and P.d.C.F.; software, B.d.S.M.G.; validation, B.d.S.M.G. and F.L.d.C.; formal analysis, B.d.S.M.G., F.L.d.C. and P.d.C.F.; investigation, B.d.S.M.G., F.L.d.C. and P.d.C.F.; resources, B.d.S.M.G., F.L.d.C. and P.d.C.F.; data curation, B.d.S.M.G.; writing—original draft preparation, B.d.S.M.G., F.L.d.C. and P.d.C.F.; writing—review and editing, B.d.S.M.G., F.L.d.C. and P.d.C.F.; visualization, B.d.S.M.G., F.L.d.C. and P.d.C.F.; supervision, F.L.d.C. and P.d.C.F.; project administration, F.L.d.C. and P.d.C.F.; funding acquisition, F.L.d.C. and P.d.C.F. All authors have read and agreed to the published version of the manuscript.

**Funding:** The APC funding was requested to São Paulo Research Foundation (FAPESP) grant number #2022/01706-3.

**Institutional Review Board Statement:** Not applicable.

**Informed Consent Statement:** Not applicable.

**Data Availability Statement:** Not applicable.

**Conflicts of Interest:** The authors declare no conflict of interest.

## Appendix A

**Table A1.** Systematization of analyzed articles.

| Authors (Year) | Title | Purpose of the Paper | Research Method | Type of Financial Indicator Used in the Study | Contributions Related to Circular Economy and Financial Aspects |
|---|---|---|---|---|---|
| Aboulamer [36] | Adopting a circular business model improves market equity value | The article aims to make a theoretical discussion regarding the possible economic benefits of adoption by circular economy practices companies. | Qualitative | The article cites indicators for determining the value of a firm and a form of determination of the value of the product for customers. However, they are presented only to explain the possible benefits. | The author understands, based on the literature, that the positive externalities of the adoption of the circular economy, perception of customers, for example, may imply financial gains (company market value). |
| Aboulamer, Soufani, and Esposito [72] | Financing the circular economic model | It debates the importance of considering the environmental impact of investments in a circular model. | Qualitative-Conceptual | Cash flows and the risk of these cash flows translated into cost of capital. | The study explains that a reliable and successful circular business model should not ignore the financial value and the risks that they represent for the value of the firm. In this sense, there is an urgent need to understand how to value new types of assets in order to enhance the allocation of funds within a circular economy. The authors state that a mismatch between the circular business model cycle and the investment horizon of some actual investors in the market represents a challenge to the allocation of more funds to the circular economy. |
| Albertario [97] | System of self-financing strategy for the policies aimed at the eco-innovation in the productive sectors | It presents a new method of theself-financing potential of an eco-oriented process system known as circular financing. | Quantitative-Modelling | Granting of funds to finance the projects of eco-innovation investment self-powered by the sum of financial and economic benefits of the system. | This study contributes with a methodology for a self-financing of eco-oriented process system—named circular financing. The methodology proposed can lead to an innovative eco-intensive economic growth that could bring the manufacturing sector to be more competitive in the long run. |

Table A1. *Cont*.

| Authors (Year) | Title | Purpose of the Paper | Research Method | Type of Financial Indicator Used in the Study | Contributions Related to Circular Economy and Financial Aspects |
|---|---|---|---|---|---|
| Alkhuzaim Zhu, and Sarkis [98] | Evaluating Emergy Analysis at the Nexus of Circular Economy and Sustainable Supply Chain Management | It evaluates various emergy analysis features and build connections between emergy, sustainable supply chain management, and circularity concepts. | Qualitative-Literature review | Emergy based performance measurement. | This article explores an effective environmentally sustainable supply chain and circularity performance evaluation, accounting, and appraisal using energy-based performance measurements. |
| Almagtome et al. [91] | Circular economy initiatives through energy accounting and sustainable energy performance under integrated reporting framework | It proposes an approach to measure the sustainable energy performance based on the integrated reporting framework. | Qualitative-Conceptual | Total eEnergy cCosts; pProduction eEnergy costs; sSaving in total energy costs; sSaving in production energy costs; eEnergy investments; rRenewable eEnergy investments. | This paper contributes by setting indicators for assessing sustainable energy performance based on the Integrated Reporting Framework. It can help regulators and economic policy makers to have accurate information on real and future of sustainable energy, as well as to develop CE strategies. |
| Almagtome, Khaghaany, and Önce [90] | Corporate governance quality, stakeholders' pressure, and sustainable development: An integrated approach | It analyzed the determinants of sustainability applications in the circular economy. | Quantitative and qualitative | Sustainable dDevelopment iIndex. | This paper found that companies with a high corporate governance record tend to disclose more economic, social and environmental information. The findings could help managers of manufacturing companies to elaborate plans to maximize the use of available resources and improve efficiency in the context of the CE. |
| Anishchenko et al. [99] | Ensuring environmental safety via waste management | It proposes a waste classification matrix for identification and recognition of waste as economic resources. | Quantitative-Modelling | Environmental taxes. | This paper recognizes the importance of developing a system to establish criteria for recognizing waste as economic resources in order to form the environmental safety management system, taking into account the main provisions of a circular economy. It is proposed that the criteria for recognition of waste as economic resources include the existence of property rights, the possibility of sale, the receipt of economic benefits and the environmental effect. |

Table A1. *Cont*.

| Authors (Year) | Title | Purpose of the Paper | Research Method | Type of Financial Indicator Used in the Study | Contributions Related to Circular Economy and Financial Aspects |
|---|---|---|---|---|---|
| Aranda-Usón et al. [92] | Measurement of the circular economy in businesses: Impact and implications for regional policies | It contributes to the measurement of the activities related to the circular economy implemented in business | Qualitative | Economic impact. | This paper discusses the economic and social impact of the circular economy, and also contributes to the development of specific regional policies to improve the circular economy in businesses. |
| Aranda-Usón, Portillo-Tarragona, Scarpellini, Llena-Macarulla, [87] | The progressive adoption of a circular economy by businesses for cleaner production: An approach from a regional study in Spain | The article analyzes how companies adopt and introduce CE principles within selected companies in Spain. | Qualitative | NA (Not applicable). | The article shows that in order to have a sustainable growth model, reducing materials for product manufacturing is needed (progressive material loops closing); environmental accounting processes and integrated indicators are needed for CE implementation. |
| Aranda-Usón, Portillo-Tarragona, Marín-Vinuesa, and Scarpellini [20] | Financial resources for the circular economy: A perspective from businesses | The main objective of this study is to define the resources applied to circular activities by companies. | Quantitative-Survey | Financial resources, rResource sSaving and efficiency. | According to the paper, the availability of funds, the quality of the firm's own financial resources, and public subsidies have a positive effect in stimulating the implementation of circular economy initiatives in businesses. |
| Bartolacci, Del Gobbo, Paolini, and Soverchia [89] | Waste management companies towards circular economy: What impacts on production costs? | The article aims to understand how separate waste collection rates affect production costs, accounting for size factors number and landed area in companies. | Qualitative and quantitative | NA (Not applicable). | The results show that separate waste collection has a positive impact on a dependent variable (in this case: production costs), and that a company's costs seem to be more influenced by size factors. |
| Bartolacci, Paolini, Quaranta, and Soverchia [100] | The relationship between good environmental practices and financial performance: Evidence from Italian waste management companies | The article aims to analyze the relationship between the financial performance of waste management companies and good environmental practices related to selective waste collection. | Quantitative | ROA. | The results presented a positive correlation between the ROA and the volume of treatment of waste per capture and between the ROA and the percentage of trash treated. |

**Table A1.** *Cont.*

| Authors (Year) | Title | Purpose of the Paper | Research Method | Type of Financial Indicator Used in the Study | Contributions Related to Circular Economy and Financial Aspects |
|---|---|---|---|---|---|
| Bockholt et al. [73] | Exploring factors affecting the financial performance of end-of-life take-back program in a discrete manufacturing context | It identifies factors that affect the financial performance of circular economy initiatives. | Qualitative-Case study | Resource efficiency: logistics and handling costs; resource effectiveness: recovered value. | This study reveals factors that from a financial standpoint should be considered when designing a take-back program for circular economy implementations. The research seeks to improve the financial viability of the circular economy, but it neglects capital investment costs and only analyses operational expenses. |
| Cong, Zhao, and Sutherland [101] | Value recovery from end-of-use products facilitated by automated dismantling planning | It proposes a two-stage dismantling planning method to improve economic performance of end-of-use products value recovery. | Quantitative-Modelling | Economic performance; value recovery. | This study explores how to circulate components/materials into another life cycle with maximal utility by proposing a method to improve the economic performance of end-of-use products. |
| Coscieme, Mortensen, Anderson, Ward, Donohue, and Sutton [102] | Going beyond Gross Domestic Product as an indicator to bring coherence to the Sustainable Development Goals | Propose guidelines for selecting alternative indicators for the 8th Sustainable Development Goal (SDG) from the United Nations, with the aim of improving coherence among all of the SDGs. | Quantitative | Gross Domestic Product (GDP), Index of Sustainable Economic Welfare (ISEW) and the Genuine Progress Indicator (GPI). | The article proposes other indicators to measure SDG 8 (Decent Work and Economic Growth), and affirms that flawed indicators measure an incoherent narrative and leads to a misleading choice. |
| Demirel, and Danisman [93] | Eco-innovation and firm growth in the circular economy: Evidence from European small- and medium-sized enterprises | This study examines the impact of circular eco-innovations and external financing available for EC activities in the growth of European SMEs. | Quantitative. | Firm growth variable, as the dependent variable, Types of CE innovations, Total investments into CE, External finance, Research and development expenditures, firm size, firm age, etc. | The study offers insights into the lower levels of SME engagement in the CE as well as policy implications for improving engagement. |
| Dewick et al. [103] | Circular economy finance: Clear winner or risky proposition? | It conducts a theoretical analysis of the risk that strategic decisions related to circular economy policy are based on imprecise and contextual indicators. | Qualitative-Conceptual. | Circular economy investments and finance. | This paper discusses that a large volume of public and private resources is needed for the transition to a circular economy. However, it is pointed out that before any investment, it is necessary to assess whether these investments are not being guided by contestable understanding, imprecise measures and inadequate information. |

Table A1. *Cont.*

| Authors (Year) | Title | Purpose of the Paper | Research Method | Type of Financial Indicator Used in the Study | Contributions Related to Circular Economy and Financial Aspects |
|---|---|---|---|---|---|
| Dey, Malesios, De, Budhwar, Chowdhury, and Cheffi [104] | Circular economy to enhance sustainability of small and medium-sized enterprises | Facilitate SMEs to achieve greater sustainability through the implementation of EC. Key strategies, resources and skills that facilitate the effective implementation of EC in SMEs. | Qualitative and quantitative | NA (Not applicable). | The current state of EC implementation in SMEs and the means of improving their absorption through strategies, resources and competencies, stakeholders play an important role in the implementation of EC. The study focuses on all EC fields and its relationship with sustainability performance. |
| Dheskali, Koutinas, and Kookos [105] | A simple and efficient model for calculating fixed capital investment and utilities consumption oflarge-scale biotransformation processes | It presents a simple and accurate mathematical model that describes the economics of the bioreaction section of a typical biotransformation technology. | Quantitative-Modelling | Fixed capital investment (FCI) | This study proposes a model which consists of just three equations that can be used to compare the economic performance of different bio-based technologies. |
| Dobrotă, Dobrotă, and Dobrescu [71] | Improvement of waste tyre recycling technology based on a new tyre marking | It seeks to identify new possibilities of superior capitalization of tyre waste to reduce environmental pollution and achieve an efficient circular economy. | Quantitative and qualitative | Net present value; Internal rate of financial return. | This research proposes a solution for enhancing the recycling process of tire waste. The financial analysis conducted shows that the technological process of capitalizing on tire waste can become an efficient one, and a circular economy for such products can be achieved. The financial sustainability of the proposed solution is verified by recording the positive cumulative cash flow. |
| Flygansvær, Dahlstrom, and Nygaard [106] | Exploring the pursuit of sustainability in reverse supply chains for electronics | The objective of this study is to understand the productive management of reverse supply chains related to electronic equipment in Norway. | Qualitative-Survey | There is not. In the study, the analysis of economic performance is measured through comparative questions regarding other companies (economic goals, growth targets etc). | The collaboration and culture between firms influence the components of Triple Bottom Line (ecological, economic and social performance).. |

**Table A1.** *Cont*.

| Authors (Year) | Title | Purpose of the Paper | Research Method | Type of Financial Indicator Used in the Study | Contributions Related to Circular Economy and Financial Aspects |
|---|---|---|---|---|---|
| Fraccascia, Yazan, Albino, and Zijm [107] | The role of redundancy in industrial symbiotic business development:A theoretical framework explored by agent-based simulation | The article proposes a combination of methodologies to support decision-making in the IS field. Affirming the effect of the redundancy strategy IS (Industrial Symbiosis) business. | Quantitative | Economic performance indicator (ECO_P); Environmental performance indicator (ENV_P). | The results point out that the weight of transaction costs are not always proportional to the environmental benefits in IS (i.e., if transaction costs are high, economic benefits generally do not simultaneously maximize the environmental benefits; and increased redundancy can lead to higher transaction costs for IS companies). |
| Gao et al. [108] | Evaluating circular economy performance based on ecological network analysis: A framework and application at city level | It aims to determine the factors that affect a city's CE performance by establishing an urban ecological network, simulating the resource utilization process of an internal urban system. | Quantitative-Modelling | Resource productivity (RP), Recycling rate (RR) and Waste disposal amount (WDA). | This research provides a method for analyzing a city's CE performance from the perspective of socioeconomic metabolism. For evaluating the effectiveness of a city's CE implementation, three indicators were selected. The authors found a strong positive correlation between urban resource productivity and the economic development level. Overall, the amounts of material input or waste discharge have important impacts on resource productivity or waste disposal, which limits urban CE performance. |

**Table A1.** *Cont.*

| Authors (Year) | Title | Purpose of the Paper | Research Method | Type of Financial Indicator Used in the Study | Contributions Related to Circular Economy and Financial Aspects |
|---|---|---|---|---|---|
| Gigli, Landi, and Germani [67] | Cost-benefit analysis of a circular economy project: a study on a recycling system for end-of-life tyres | The aim of the paper is to illustrate an innovative technology for ELT fibre's recycling, which allows for the transforming of textile fibre into a useful secondary raw material for different applications. | Case study | Economic Net Present Value (ENPV), Economic Rate of Return (ERR) and Benefit/Cost ratio (B/C ratio). | The study shows an impact reduction in case the ELT fibre is reused as an additive for bituminous conglomerates, instead of disposing of it. The financial and economic sustainability of the related technological process was evaluated to check whether the process is sustainable in the long term. According to the paper that presents the cost-benefit analysis, in the medium and long term, the system is financially viable, and the high economic profitability makes the process economically sustainable. Furthermore, sensitivity analysis, as well as a risk assessment, has been carried out in order to identify critical variables, evaluate risks and define risk mitigation measures concluding that the project is not highly risky since even in the worst scenario the possible loss is moderate. Based on the results pointed out by the authors, it can be concluded that this ELT fibre's recycling system can be replicated across Europe. |
| Gimeno, Llera-Sastresa, and Scarpellini [109] | A heuristic approach to the decision-making process of energy prosumers in a circular economy | It explores insights into the decision-making process of energy prosumers to enhance the understanding of self-consumption deployment and to support effective policymaking. | Qualitative-Case study | Payback; Access to financing; Installation costs. | This article provides a heuristic analysis of investment decision-making related to sustainable energy at a small scale, using the economic criteria of installation costs or energy consumption data for the calculation of payback. The results showed that contextual factors influencing the final decisions were mainly related to the investment return and the future performance of the installation in both economic and technical aspects. |



**Table A1.** *Cont.*

| Authors (Year) | Title | Purpose of the Paper | Research Method | Type of Financial Indicator Used in the Study | Contributions Related to Circular Economy and Financial Aspects |
|---|---|---|---|---|---|
| Gu et al. [110] | Environmental performance analysis on resource multiple-life-cycle recycling system: Evidence from waste pet bottles in China | It builds an environmental performance measurement method based on the multiple life cycle recycling (MLCA) process and simulates the general impact of various policies on the PET bottle recycling system. | Quantitative-Modelling | Tons of waste. | This study suggests to replace the life cycle assessment (LCA) by multiple life cycle assessment (MLCA). The authors point out that the effect of saving resources and reducing environmental emissions in the recycling process must be accurately accounted for. They suggest the need for a top-level, multi-departmental natural resource management organization to be established as quickly as possible, with the aim of maximizing the overarching environmental and economic benefits of MLC processes. |
| Haas, Krausmann, Wiedenhofer, and Heinz [65] | How circular is the global economy?: An assessment of material flows, waste production, and recycling in the European union and the world in 2005 | The research applies a sociometabolic approach to assess the circularity of global material flows. | Quantitative and Qualitative | Efficiency. | The result shows that the European Union has a slightly higher index than the rest of the world with regard to recycling and the circularity of materials. The study affirms the need for an eco-design (Sustainable Design) for the circular economy to advance, resulting in renewable energy, recycling, and product design. |
| Hald, Wiik, and Larssen [111] | Sustainable procurement initiatives and their risk-related costs: a framework and a case study application | It develops and applies a framework designed to identify and measure the risk-related cost trade-offs inherent in initiatives designed to improve sustainability in procurement. | Qualitative-Case study | Individual expected monetary value (EMV). | This article contributes to the existing literature by developing a framework to outlining the risk-related cost implications of investments initiatives designed to improve sustainability in procurement and to achieve a circular economy. |
| Hao et al. [112] | Modeling and techno-economic analysis of a novel trans-critical carbon dioxide energy storage system based on life cycle cost method | It proposes a techno-economic model to assess the investment costs and economic performance by using the life cycle cost method on CO2 utilization of the energy storage system. | Quantitative-Modelling | Production operating costs, converted energy benefits and financial performance indicators (NPV). | This article analyzes the investment cost and economic performance of an energy storage model. |

Table A1. *Cont.*

| Authors (Year) | Title | Purpose of the Paper | Research Method | Type of Financial Indicator Used in the Study | Contributions Related to Circular Economy and Financial Aspects |
|---|---|---|---|---|---|
| Helander, Petit-Boix, Leipold, and Bringezu [68] | How to monitor environmental pressures of a circular economy | The article objectives to assess whether and to what extent current approaches to assessing CE activities sufficiently capture environmental pressures to monitor progress toward environmental sustainability. | Qualitative-Conceptual | The article analyzes the preexisting indicators, placing the purpose of each of the identified in context. | The paper shows, based on a material flow perspective, that most indicators do not capture environmental pressures related to the CE activities they address. The study suggests complementing CE management indicators with indicators capturing basic environmental pressures related to the respective CE activity. |
| Hens, Block, Cabello-Eras, Sagastume-Gutierez, Garcia-Lorenzo, Chamorro, Herrera Mendoza, Haeseldonckx, Vandecasteele [113] | On the evolution of "Cleaner Production" as a concept and a practice | This paper provides a review of essentials that contributed to the fundamental changes in Cleaner Production during the most recent quarter of a century, and the concepts it contemplates, including the circular economy, corporate social responsibility, environment, Green Smart Cities, sustainable tourism, etc. | Literature review | The study finds that there is no widely accepted indicator that enables monitoring of an organization or a larger unit. | The conclusion is that the cost is fundamental in green accounting for cleaner production. The implementation of an environmental accounting system is more relevant as it benefits society and entrepreneurs. It contributes to the appropriate management of natural resources and the environment, a fundamental objective of green accounting. |
| Ibrahim, and Shirazi [114] | The role of Slamic finance in fostering circular business investments: The case of OIC countries | It explores the role of Islamic finance in fostering investments towards the CE to optimize resource use and avoid waste in the course of economic growth. | Qualitative | Financing and investment tools—Blended finance. | The work states that blended finance is a valuable tool that can be used to foster investment and attract funding for circular businesses. The authors explain that the nature of circular economic growth is that it is perceived risky by investors. In this sense, only commercial capital cannot reach the required impact and hence the need to blend commercial capital, charitable capital, and public capital. They conclude that the Islamic finance can use compassionate contracts, equity-like, and risk-sharing financing modes to support circular businesses. |

Table A1. *Cont*.

| Authors (Year) | Title | Purpose of the Paper | Research Method | Type of Financial Indicator Used in the Study | Contributions Related to Circular Economy and Financial Aspects |
|---|---|---|---|---|---|
| Ionaşcu, and Ionaşcu [115] | Business models for circular economy and sustainable development: The case of lease transactions | It discusses the characteristics of leasing as a business model in the circular economy, which is assumed to support sustainable development through product recirculation and boosting economic performance. | Quantitative. | Return on assets, return on sales, q de Tobin, market value, and market to book. | This article shows that financial performance is generally higher for Romanian listed companies that use leasing and renting, and that performance is also directly associated with the intensity of the leasing. |
| Jain, Panda, and Choudhary [116] | Institutional pressures and circular economy performance: The role of environmental management systemand organizational flexibility in oil and gas sector | It investigates whether institutional pressures influence environmental and economic performance towards the circular economy. Based on institutional theory, it examines the role of organizational flexibility on the effects of institutional pressures on circular economy performance through environmental management systems (EMS). | Quantitative-Survey | Economic performance. | The study reveals that organizations leverage the environmental management system to achieve CE performance, and flexible organizations, compared to rigid ones, are more effective in dealing with coercive pressures when leveraging the environmental management system. |
| Kerdlap, Low, and Ramakrishna [117] | Life cycle environmental and economic assessment of industrial symbiosis networks: a review of the past decade of models and computational methods through a multi-level analysis lens | It examines the state-of-the-art methodologies used in life cycle assessment and life cycle costing (LCC) of Industrial symbiosis networks. | Qualitative-Literature review | Financial feasibility, net present value, profitability. | This study found that existing LCC methodology is successful in modeling all capital and operation costs, taxes, and waste treatment and recycling costs, as well as the time value of money to determine the feasibility, profitability, NPV, and other economic aspects of industrial symbiosis networks. |
| Khan, and Badjie [118] | Islamic blended finance for circular economy impactful SMEs to achieve SDGs | It presents a framework for blended Islamic finance for impactful small and medium enterprises to implement the circular economy. | Qualitative | Financing and investment tools—Blended finance. | The study proposes a structure to create a win–win result for the blending parties, considering that the blended nature of the designed contract provides a social subsidy to fund the cost element of the financing. It is explained that the collaborative and innovative proposed contract design can contribute to achieving multidimensional human development, the circular economy, and the sustainable development goals. |

Table A1. *Cont*.

| Authors (Year) | Title | Purpose of the Paper | Research Method | Type of Financial Indicator Used in the Study | Contributions Related to Circular Economy and Financial Aspects |
|---|---|---|---|---|---|
| Kimata, and Itakura [119] | Interactions between organizational culture, capability, and performance in the technological aspect of society: Empirical research into the Japanese service industry | It investigates the way of balancing environmental protection with corporate profits. | Quantitative-Survey | Economic performance. | This article reveals that a sustainable balance between the environment and economics is easily achieved when the level of organizational capability is high, which means that efforts to improve the environmental protection above certain levels might be unprofitable for firms with a low level of organizational capability. Therefore, a balance between the environment and economics is achieved through a strong environmental protection culture especially based on values. |
| Kuo, Chiu, Chung, and Yang [120] | The circular economy of LCD panel shipping in a packaging logistics system | It analyzed the packaging logistics system in LCD panel manufacturing and compared the cost of traditional logistics mode and green logistics mode to identify the economic performance. | Quantitative-Simulation | Economic performance. | This article's results show that the total cost of the green logistics model is beneficial and always lower than the traditional logistics model, even with a low recycling rate. |
| Larsen, Masi, Feibert, and Jacobsen [121] | How the reverse supply chain impacts the firm's financial performance: A manufacturer's perspective | Identifying how the reverse supply chain can contribute to the financial performance of companies. | Qualitative and Quantitative | NA (Not applicable). | The article finds in the literature 15 functions that try to explain how the reverse supply chain can impact the financial performance of the companies. |
| Li et al. [85] | Resource value flow analysis of paper-making enterprises: A Chinese case study | It analyzes the value stream related to the use of resources from a circular economy perspective, developing an extension of material flow cost accounting and modifying it by accounting for environmental damage as well as economic benefits. | Qualitative- Case study | Costs in the production flow, considering the costs of external environmental damage. | The article analyzes the resource value stream through an analysis of the relationship between material flows and value flows. For this, it evaluates the cost flow of the production process considering the internal resources lost in external environmental damage costs. |

Table A1. *Cont.*

| Authors (Year) | Title | Purpose of the Paper | Research Method | Type of Financial Indicator Used in the Study | Contributions Related to Circular Economy and Financial Aspects |
|---|---|---|---|---|---|
| Li et al. [122] | Green innovation and business sustainability: new evidence from energy intensive industry in China | It proposes to test the relationship between green innovation practices and business sustainability among energy-intensive Chinese companies. | Quantitative-Survey | Financial performance. | This paper showed that green innovation has three dimensions: green product innovation, recycling and green advertising. Corporate sustainability also had three dimensions: financial performance, environmental performance and social performance. It also found that green innovation has a significant effect on business sustainability in the energy-intensive industry. |
| Llanquileo-Melgarejo et al. [123] | Evaluation of the impact of separative collection and recycling of municipal solid waste on performance: An empirical application for Chile | It evaluates the impact of selective collection and recycling of municipal solid waste (MSW) on the performance of municipalities in providing MSW services. | Quantitative-Data envelopment analysis | Total costs of MSW collection and disposal. | This work shows that the selective collection and recycling of MSW has an impact on the performance of municipalities, affecting their abilities to achieve a circular economy. |
| Maranesi, and De Giovanni [124] | Modern circular economy: Corporate strategy, supply chain, and industrial symbiosis | It analyzes the chances of companies considering the circular economy as part of their corporate strategy. | Qualitative-Case study | Efficiency. | This paper concluded that the circular economy is a business accelerator, as it provides the opportunity for companies to improve their environmental impact and social contribution, as well as discover new and atypical business opportunities, involving senior management and shareholders, supply chain members, industrial partners and consumers. |

**Table A1.** *Cont.*

| Authors (Year) | Title | Purpose of the Paper | Research Method | Type of Financial Indicator Used in the Study | Contributions Related to Circular Economy and Financial Aspects |
|---|---|---|---|---|---|
| Marrone et al. [125] | Trends in environmental accounting research within and outside of the accounting discipline | It assesses the emergence of research topics and trends in environmental accounting. | Qualitative-Literature review | Environmental accounting. | This paper indicated that research studies in accounting journals have addressed sustainability issues in a general way, with a recent focus on broad topics such as corporate social responsibility (CSR) and stakeholder theory. Research studies published outside accounting journals have focused on more specific topics (e.g., moving to a circular or low-carbon economy, meeting sustainable development goals [SDGs], etc.) and new methodologies (e.g., accounting for ecosystem services). |
| Millward-Hopkins, and Purnell [126] | Circulating blame in the circular economy: The case of wood-waste biofuels and coal ash | The article discusses if wood residues used for power generation may in fact be considered residue. In addition, it discusses a form of carbon credit measurement due to the use of wood waste and how the industry can influence this practice. | Qualitative-Case study | NA (Not applicable). | The conclusion of the article punctuates the need for evolution of policies and responsibilities associated with carbon emissions, and how these policies can influence the agendas of circular and low carbon economy. |
| Naims [127] | Economic aspirations connected to innovations in carbon capture and utilization value chains | It investigates the economic expectations placed on those actors currently conducting research and development (R&D) in carbon capture and utilization (CCU). | Qualitative-Literature review | Economic performance. | This study has shown how CCU innovations have different economic impacts on multiple industries along a value chain. |

**Table A1.** *Cont.*

| Authors (Year) | Title | Purpose of the Paper | Research Method | Type of Financial Indicator Used in the Study | Contributions Related to Circular Economy and Financial Aspects |
|---|---|---|---|---|---|
| Ngan, How, Teng, Promentilla, Yatim, Er, and Lam [128] | Prioritization of sustainability indicators for promoting the circular economy: The case of developing countries | The work aims to provide a comprehensive review of the circular economy concept in developing country context and a novel model is proposed by adopting Fuzzy Analytics Network Process (FANP). | Case study and Fuzzy Analytic Network Process (FANP) | Economic: Cost and Profit; Environmental: Carbon footprint, Water footprint, and Ecological; Social: Health and Safety, Education and training, and Public acceptance. | The results show that economic performance indicators still play a dominant role in encouraging industry actors to adopt sustainable practices to promote CE. This indicates that economic benefits and public acceptance play a prominent role in affecting the decision of industry participants in relation to the CE. Local authorities need to adopt the recommendations to design policy and incentives that encourage the adoption of CE in real industry operation to spur up economic development, without neglecting environmental well-being and jeopardizing social benefits. |
| Pamfilie et al. [129] | Circular economy—A new direction for the sustainability of the hotel industry in Romania? | It studies the influence of the implementation of integrated quality-environment-safety systems (ISO 9001, 14001, UHSAS 18001) on the economic performance of hotels in Romania. | Quantitative-Survey | RevPAR (a performance indicator in the hotel industry used to analyze investment decisions). | This study concludes that the Romanian hotel industry is not sufficiently prepared to adopt the circular economy principle, and that the adoption of an integrated management system is not having as much influence as believed on the performances of the operators in the field. |
| Pauliuk [66] | Critical appraisal of the circular economy standard BS 8001:2017 and a dashboard of quantitative system indicators for its implementation in organizations | The study evaluates a technical standardization of circular economics for companies, elaborated by the British Standards Institution, which reconciles the CE long-range ambitions with established business routines. | Quantitative and Qualitative | Circular economy index (CEI) and Ratio of recirculated economic value from EoL (End of Life). | The study points out that monitoring and strategy of the circular economy were not met by the evaluated technical standardization. It also states that, in the perspective of the life cycle of materials, central indicators of the circular economy should refer to the depletion of natural resources, the growth of stock in use, and the life of the materials. The author also points out that inconsistent indicators on circularity can be a risk to companies that adopt the circular economy. |

**Table A1.** *Cont*.

| Authors (Year) | Title | Purpose of the Paper | Research Method | Type of Financial Indicator Used in the Study | Contributions Related to Circular Economy and Financial Aspects |
|---|---|---|---|---|---|
| Portillo-Tarragona et al. [86] | Classification and Measurement of the Firms' Resources and Capabilities Applied to Eco-Innovation Projects from a Resource-Based View Perspective | Define and measure the specific resources applied to investments in eco-innovation by companies and analyze its financial and environmental influences eco-inovation projects. | Qualitative and quantitative | Profitability Index, IRR, INV, PayBack, ROE, % investment in research and development, among others. | The study provides definitions and classifications of indicators specifically designed for eco-innovation projects. In the projects analyzed, results regarding the economic-financial resources and capabilities were slightly predominant. |
| Rehman Khan et al. [130] | The role of blockchain technology in circular economy practices to improve organisational performance | It examines the role of blockchain technology for the circular economy to improve organizational performance in the context of the China-Pakistan Economic Corridor (CPEC) | Quantitative-Survey | Economic performance. | This paper demonstrates that blockchain technology plays a positive role in the circular economy, which leads to a positive nexus with environmental and economic performance, also boosting organisational performance. |
| Rieckhof and Guenther [95] | Integrating life cycle assessment and material flow cost accounting to account for resource productivity and economic-environmental performance | The purpose of the article is to apply the existing knowledge regarding the life cycle assessment and its relationship with the accounting of material flow costs. | Qualitative-Case Study | NA (Not applicable). | When comparing critical resource use points, as well as associated costs and environmental charges, instruments, together, provide valuable perceptions to identify the integrated resource economy, as well as the potential economic and environmental improvements for business operations current and future. |
| Santagata, Zucaro, Viglia, Ripa, Tian, and Ulgiati [131] | Assessing the sustainability of urban eco-systems through Emergy-based circular economy indicators | To design a reasonable CE structure, a series of existing and innovative processes are analyzed and discussed. Investigate CE development options based on locally recovering resource recovery and still usable (residues). | Qualitative | NA (Not applicable). | The study provides a structure for energy flows and materials towards assets in order to implement CE, besides effective sustainable paths for urban ecosystems. |
| Scarpellini et al. [23] | Dynamic capabilities and environmental accounting for the circular economy in businesses | It defines and measures the environmental capabilities that are applied when circular economy concepts are introduced in a business. | Quantitative | Return on equity, return on sales, return on assets. | This paper suggests the existence of a positive relationship between the circular scope of companies and their practices of environmental accounting, corporate social responsibility (CSR), and accountability. |

**Table A1.** *Cont.*

| Authors (Year) | Title | Purpose of the Paper | Research Method | Type of Financial Indicator Used in the Study | Contributions Related to Circular Economy and Financial Aspects |
|---|---|---|---|---|---|
| Scarpellini, Marín-Vinuesa, Portillo-Tarragona, and Moneva [70] | Defining and measuring different dimensions of financial resources for business eco-innovation and the influence of the firms' capabilities | The objective of this study is to define, classify and measure the different dimensions of the financial resources applied to the eco-innovation by companies and to analyze the influence of the technological and environmental management capabilities of companies in the efficient allocation of these resources to carry out investments in eco-innovation. | Survey and Partial Least Square Structural Equation Model (PLS-SEM) | Information related to eco-innovation level, Financial resource quality, Public financial resources, Financial Resources Availability, Technological and sectorial capabilities, Environmental management capabilities and Firm Size. | The paper shows that different dimensions of financial resources influence the eco-innovative investment and the internal management of eco-innovation. |
| Secondi [132] | A regression-adjustment approach with control-function for estimating economic benefits of targeted circular | It seeks to estimate the average effects of various circular economy practices on small and medium-sized Enterprises' economic performance. | Quantitative-Survey | NA (Not applicable). | The paper shows the importance of intensifying and accelerating the transition to circularity, especially focusing on the optimization of waste management and the use of renewable energy. |
| Susanty, Tjahjono, and Sulistyani [133] | An investigation into circular economy practices in the traditional wooden furniture industry | It analyzes how the different levels of environmental-oriented supply chain cooperation (ESCC) practices impact CE environmental and economic performance. | Quantitative-Survey | Economic performance indicators: increase in percentage of income due to activities from wood waste management; decrease in percentage of cost due to purchase a new wood; decrease in percentage of costs due to defective furniture component/material; decrease in percentage of costs due to excess wood inventory; decrease in percentage of costs due to wood waste disposal. | This paper shows that CE's economic performance indicates the ability of the company to reduce costs related with energy consumption, purchased raw materials, fines for environmental accidents, and waste discharge/treatment. The results revealed a positive correlation between CE economic and environmental performance and the implementation of ESCC practices. |

**Table A1.** *Cont.*

| Authors (Year) | Title | Purpose of the Paper | Research Method | Type of Financial Indicator Used in the Study | Contributions Related to Circular Economy and Financial Aspects |
|---|---|---|---|---|---|
| Svensson, and Funck, [74] | Management control in circular economy. Exploring and theorizing the adaptation of management control | How organizations work with the circular economy and how management control has adapted to the business model. | Qualitative | NA (Not applicable). | The results show that the circular economy can be practiced in different ways and that the adaptation and application of the circular economy affects the entire management control package. Action plans, cost accounting and investment assessments should reflect a higher level of details and a longer time horizon. |
| Taleb, and Al Farooque [134] | Towards a circular economy for sustainable development: An application of full cost accounting tomunicipal waste recyclables | It analyzes different accounting approaches and scenarios for the sustainable management of municipal waste, seeking to identify the most economical and profitable approach. | Quantitative-Modelling | Fixed costs (investments) and variable costs (associated with garbage collection and treatment, as well as the possible benefits generated by collection, both direct (recycling) and indirect (environmental benefits). | This study found that the prepaid bag system in the method volume-based PAYT leads to lower waste costs and creates more incentives for families in terms of economic, social and environmental benefits towards a CE. |
| Titova [135] | Principles of circular economy introduction in Russian industry | It explores the principles that enable the introduction of a circular economy in Russian industry. | Qualitative-Content analysis | Accounting and assessment of the potential cost of waste: total cost of waste; differentiation of waste in accordance with its potential value; profit from waste use Resource differentiation: the cost of primary resource replacement; the price ratio between primary and secondary resources; minimization of production costs Saving on renewable resources: volume of non-renewable resource consumption; savings from the use of renewable resources Increased product life: product replacement cost. | This work explains the key principles for introducing a circular economy in industries, which are the principle of accounting and assessing the potential cost of waste, the principle of resource differentiation, the principle of saving from the use of renewable resources, and the principle of a product life increase. |

**Table A1.** *Cont.*

| Authors (Year) | Title | Purpose of the Paper | Research Method | Type of Financial Indicator Used in the Study | Contributions Related to Circular Economy and Financial Aspects |
|---|---|---|---|---|---|
| Tuccio et al. [136] | Mass balance as economic and sustainable strategy in WEEE sector | It investigates environmental and economic dimensions in the e-waste secto. | Qualitative-Case study | Waste recovery tax. | This study used a novel approach of mass balance analysis, which is an approach capable of linking economic and ecological dimensions, to evaluate eco-efficiency in the e-waste sector as a way to produce secondary raw material for a CE model. |
| Vegera, Malei, Sapeha, and Sushko [137] | Information support of the circular economy: The objects of accounting at recycling technological cycle stages of industrial waste | The purpose of this study is to distinguish the technological cycle stages of industrial waste recycling and to identify the accounting objects at these stages. | Analysis and synthesis, comparison, logical generalization, inference by analogy, classification, grouping, etc | The authors portray quality indicators, quoting other authors. | In the article, technological cycle stages of industrial waste were considered and were identified as the objects of their accounting. |
| Wang, Che, Fan, and Gu [138] | Ownership Governance, Institutional Pressures and Circular Economy Accounting Information Disclosure—an institutional theory and corporate governance theory perspective | This paper aims to explore the determinants of circular economy accounting information disclosure quality, and also to make empirical analysis on the relationship between circular economy accounting information disclosure quality and corporate ownership governance and institutional pressures according to institutional theory and corporate governance theory. | Quantitative | Circular economy accounting information disclosure index, ownership concentration ratio, and financial ratios. | The results suggest that ownership governance and institutional pressures mainly determine the quality of circular economy accounting information disclosure. |

**Table A1.** *Cont.*

| Authors (Year) | Title | Purpose of the Paper | Research Method | Type of Financial Indicator Used in the Study | Contributions Related to Circular Economy and Financial Aspects |
|---|---|---|---|---|---|
| Whicher, Harris, Beverley, and Swiatek [139] | Design for circular economy: Developing an action plan for Scotland | This article explores the question of how design-oriented innovation can be incorporated into the action plans of the regional circular economy. | Design process | Economic and social benefits from the circular economy. | The paper reports a project undertaken in Scotland to developed policy proposals, aligning market and government needs in order to create favorable conditions for the public and private sector to adopt circular principles. The study summarizes a number of good practices drawn from this experience that may be used, according to the authors, in other countries that are looking to develop a circular economy policy framework. |
| Yang et al. [140] | Optimization of circular economy of large-scale pig farm based on material flow cost accounting | It analyzes the whole cost of large-scale pig farm based on the principle of material flow cost accounting. | Quantitative-Modelling | Material flow cost accounting (MFCA). | This paper uses the material flow cost accounting (MFCA) to determine the loss of resources and to propose optimization suggestions of circular economy for large-scale pig farms. |
| Yin, Wang, Zhou, and Liang [96] | Review of eco-efficiency accounting method and its applications (Review) | This paper proposes a literature review about Eco-efficiency accounting methods and its applications in different scales. | Qualitative | NA (Not applicable). | Eco-efficiency methods are present in accounting and financial areas, as well as multi-index systems, although some seem to be more present in an international context. |
| Zacho, Mosgaard, and Riisgaard [69] | Capturing uncaptured values—A Danish case study on municipal preparation for reuse and recycling of waste | The study intends to provide more detailed knowledge about the size and characteristics of the potential value to be captured from resources embedded in waste. | Case study | Cost-effectiveness. | The results suggest that the current regulation of the waste sector does not sufficiently support a transition to the circular economy. From a focus on cost-effectiveness, waste managers should additionally change their attention toward a focus on value creation and increase activities related to reuse. |
| Zamfir, Mocanu and Grigorescu [88] | Circular economy and decision models among European SMEs | The paper explores entrepreneurial decision models for adopting circular economy practices, focusing on European SMEs. | Quantitative | Level and nature of investments and economic performances of companies. | The main findings of the article offer a better understanding of the relationship between characteristics of European SMEs and their decisions in the field of the circular economy. |

**Table A1.** *Cont.*

| Authors (Year) | Title | Purpose of the Paper | Research Method | Type of Financial Indicator Used in the Study | Contributions Related to Circular Economy and Financial Aspects |
|---|---|---|---|---|---|
| Zhou, Zhao, Chen, and Zeng [84] | MFCA extension from a circular economy perspective: Model modifications and case study | The study combines the traditional Environmental Management Accounting (MFCA) model with the circular economy (CE), using an iron and steel industry as an example, and examines how to modify the MFCA model for an iron and steel enterprise according to the perspective of CE. | Case study | Material Flow Cost Accounting (MFCA). | The connection between material flow accounting and current accounting systems is the key to the application of the MFCA. The modified MFCA combines and innovates the CE resource flow analysis method, value flow accounting procedures, flow centers division patterns, standard cost control and adjustment, and innovation technology. Based on the management of the PDCA cycle, the modified MFCA model collects the essential functions on data processing, value accounting, information disclosure, and evaluation analysis. This optimizes decisions related to resource flow values to build a unique application model for CE decision-making in iron and steel companies. |

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
