# Peer review of "Circular Economy and Financial Aspects: A Systematic Review of the Literature"

_sustainability, doi:10.3390/su14053023_

Round 1
Reviewer 1 Report
I find the article interesting and properly structured. Moreover, the subject matter is contemporary and relevant to the economy of the 21st century. The analysis is also well-constructed and the conlusions are properly formulated. However, I would like to suggest taking into account the results presented in the following publications, showing, that European researchers also are interested in the circural economy problems:
- Batyk, I.M.; Farelnik, E.; Rakowska, J.; Maciejczak, M. Polish Cittaslow Local Governments’ Support for Renewable Energy Deployment vs. Slow City Concept. Energies 2022, 15, 201. https://doi.org/10.3390/en15010201
- Drejerska Nina, Vrontis Demetris, Siachou Evangelia [et al.], System solutions for the circular economy on the regional level: the case of Green Lungs of Poland. Journal for Global Business Advancement, 2020, vol. 13, no. 4, pp.447-458. DOI:10.1504/JGBA.2020.10034559
- Renato L Binati, Elisa Salvetti, Anna Bzducha-Wróbel, Loreta BašinskienÄ—, Dalia ÄŒižeikienÄ—, David Bolzonella, Giovanna E Felis, Non-conventional yeasts for food and additives production in a circular economy perspective, FEMS Yeast Research, Volume 21, Issue 7, November 2021, foab052, https://doi.org/10.1093/femsyr/foab052
- From a concept to implementation of food chain within the Circular Economy paradigm : the case of Poland, in [Rivista di studi sulla sostenibilità : VIII, 1, 2018][Milano : Franco Angeli, 2018.] - Permalink: http://digital.casalini.it/10.3280/RISS2018-001007
I hope that the proposed changes would increase the interest in the publication among European researchers, which would be beneficial for the journal and for the authors of this publication..
Author Response
Manuscript ID sustainability-1557257
Special Issue: Sustainable Finance and Value Creation
“Circular Economy and Financial Aspects: A Systematic Review of the Literature”
Dear Editor,
We have gone through the valuable comments provided by the reviewers and have substantially revised our manuscript. Please, see the additions highlighted in the manuscript using Track Changes/Comments tools of Word. We are sending two versions of our manuscript, the first version with changes highlighted and another with the changes accepted on the text.
We are thankful to the anonymous reviewers for their insightful comments. As you can see, their comments have improved the quality of the paper significantly.
As you review our point-by-point response to reviewer comments attached to this letter, we are confident that you will agree that we have addressed their comments satisfactorily.
We are thankful to you for giving us a chance to overhaul this paper. Please, see the enclosed itemized reply to reviewers.
With Kind Regards,
The Authors
January 30, 2022
Response to Reviewers’ Comments
Reviewer #1
|
Suggestion |
Reply |
|
General comment: “I find the article interesting and properly structured. Moreover, the subject matter is contemporary and relevant to the economy of the 21st century. The analysis is also well-constructed and the conclusions are properly formulated.” |
Dear Reviewer 1, thank you for the important suggestion below and for considering the relevance of our research.
|
|
Suggestion 1: However, I would like to suggest taking into account the results presented in the following publications, showing, that European researchers also are interested in the circural economy problems:
1. Batyk, I.M.; Farelnik, E.; Rakowska, J.; Maciejczak, M. Polish Cittaslow Local Governments’ Support for Renewable Energy Deployment vs. Slow City Concept. Energies 2022, 15, 201. https://doi.org/10.3390/en15010201 2. Drejerska Nina, Vrontis Demetris, Siachou Evangelia [et al.], System solutions for the circular economy on the regional level: the case of Green Lungs of Poland. Journal for Global Business Advancement, 2020, vol. 13, no. 4, pp.447-458. DOI:10.1504/JGBA.2020.10034559 3. Renato L Binati, Elisa Salvetti, Anna Bzducha-Wróbel, Loreta BašinskienÄ—, Dalia ÄŒižeikienÄ—, David Bolzonella, Giovanna E Felis, Non-conventional yeasts for food and additives production in a circular economy perspective, FEMS Yeast Research, Volume 21, Issue 7, November 2021, foab052, https://doi.org/10.1093/femsyr/foab052 4. From a concept to implementation of food chain within the Circular Economy paradigm : the case of Poland, in [Rivista di studi sulla sostenibilità : VIII, 1, 2018][Milano : Franco Angeli, 2018.] - Permalink: http://digital.casalini.it/10.3280/RISS2018-001007 I hope that the proposed changes would increase the interest in the publication among European researchers, which would be beneficial for the journal and for the authors of this publication.” |
Dear Reviewer 1, thank you for the suggestion. We have considered in Theoretical Foundation the suggested articles, but unfortunately, we couldn’t access papers 3 and 4. We have also reaffirmed Europe's importance in Circular Economy context. |
Reviewer #2
|
Suggestion |
Reply |
|
General comment: “The manuscript suggests a current and attractive topic for academia. The effort made is evident, but it presents conceptual errors in the methodology, an inadequate search for terms and poor analysis, which do not allow it to meet the standards of bibliometric studies.” |
Dear Reviewer 2, thank you for all the valuable suggestions and for recognizing our efforts in developing this study. We hope to have addressed all your comments satisfactorily.
|
|
Suggestion 1: “Abstract: This section needs to be rewritten. The abstract should contain concisely in a single paragraph the importance of the study, the objective, methodology, results, discussion and conclusions.” |
Dear Reviewer, thanks for this comment. We have rewritten the abstract to highlight the importance of the study, the objective, methodology, results, discussion, and conclusions. Please, see the new abstract in the manuscript. |
|
Suggestion 2: “Introduction: In general terms, should revise the introduction. The literature on Circular Economy and Financial Performance is very poor. It shows few references and a lot of unsubstantiated text, which is incompatible for a prestigious journal. Moreover, the few references used do not show authors of reference in the study area.” |
Dear reviewer, thank you for the recommendations for improving the introduction. The introduction section was revised, and we did our best to include all your comments. In this sense, we added relevant references in the CE area, clearly stated the objective of the paper, its importance as well as the structure of the paper. Please, see the additions in the manuscript. |
|
Suggestion 3: “Introduction: It does not state the study's objective (it should indicate the word aim).” |
As suggested, we have now clearly stated the objective of the paper, using the word “aim”. |
|
Suggestion 4: “Introduction: It should indicate the importance of the study for the academy. In other words, a bibliometric study.” |
Dear reviewer, we have better positioned our study considering the existing literature about CE and bibliometric method, and also explained how our study brings different contributions from the other papers. |
|
Suggestion 5: “Introduction: Do not mention relevant studies in the area ("circular economy", "bibliometric"). For example Worldwide research on circular economy and environment: A bibliometric analysis (Ruiz-Real et al. 2018), research insights (Goyal et al., 2021), Circular Economy. A review and bibliometric analysis (Camón & Celma), The circular economy umbrella: Trends and gaps on integrating pathways ( Homrich & Galvao, 2018). Or together ("Circular economy & performance") Nexus of circular economy and sustainable business performance in the era of digitalization (Agrawal et al., 2021), Circular economy performance assessment methods: A systematic literature review (Sassanelli et al., 2019). Regarding the last two references, what is special about the study to make it relevant when similar studies exist?.” |
Thank you for pointing out these important studies. We included the suggested references, which enhanced our paper by helping to position our study considering the existing literature about CE and bibliometric method. In this sense, as mentioned in Suggestion 4, we have clearly stated how our study brings different contributions from the other papers. Please, see the last three paragraphs of the new introduction. |
|
Suggestion 6: “Introduction: The authors need to state how the article is going to be divided (number of sections) and what each section comprises.” |
Thank you for pointing out this problem. We included a paragraph to explain the structure of the paper. Please, see the last paragraph of the introduction. |
|
Suggestion 7: “Theoretical Foundation: There is an interesting section on the circular economy (from a historical point of view) rather than on business application (the latter should be the central point due to the subject matter). The curious thing is that "financial performance" (the second important term), which is mentioned in the title, does not appear in this section or is not mentioned at all. Therefore, the authors should agree on the content of the body or the title because it is not related.” |
Dear Reviewer, thank you for the suggestion. We have renamed the Title to align in a better way the title with the objective and the results of the paper. |
|
Suggestion 8: “Materials and Methods: This section requires complete restructuring, expansion and new references (there are practically no references for a modern and fashionable topic). Bibliometric studies have a fundamental characteristic; they must be systematic, transparent and reproducible. As far as possible, a sequence of methodological stages (steps) and a figure summarising the process should be indicated.” |
Dear reviewer, thank you for pointing out this issue. To meet your recommendation, we have restructured our whole Materials and Methods section. Please, see the new structure of the Materials and Methods section in the manuscript. |
|
Suggestion 9: “Materials and Methods: Lines 177 - 178. The above definition is almost the classic definition of Alan Pritchard from the 1960s. Use more renowned references.” |
Dear reviewer, thank you for the suggestion. Our whole Materials and Methods section was restructured including new and important references, the paper from Groos and Pritchard (1969) was included. Please see section 3. |
|
Suggestion 10: “Materials and Methods: Line 176 indicates two divisions. This is conceptually flawed. Bibliometric analyses are divided into performance and mapping. See Cobo et al. 2011 "An approach for detecting, quantifying, and visualizing the evolution of a research field: A practical application to the Fuzzy Sets Theory field".” |
Dear Reviewer, thank you for the suggestion and reference indication. We have restructured the Materials and Method to make it clearer. Please see section 3. |
|
Suggestion 11: “Materials and Methods: Why use Scopus? There are other important databases such as Web of science and Dimensions. Use arguments and references to defend your idea. Do not use the argument of Scopus itself; it must be of scientific articles.” |
Dear reviewer, we have further clarified our choice of using Scopus database. Please, see the new arguments in the Section 3.1.
“For finding these studies, the database chosen was Scopus (Elsevier). The Scopus platform was selected because of its vast collection with more than 73 million records in 24 thousand journals (Elsevier, 2021). Besides that, Scopus is one of the most used databases in the academic world (Kipper et al., 2019; Geraldi et al., 2011; Harzing and Alakangas, 2016).” |
|
Suggestion 12: “Materials and Methods: Lines 230 - 232. The search is questionable. The title is about "financial performance" but is not used here. It adds terms "Finance”, "Accounting", which are not necessarily related to performance. This search is questionable and unreliable.” |
Dear reviewer, thank you for the observation. It is important to point out that the search is not restricted to “circular economy” as the first search term and “Finance OR Accounting” as the second, but “circular economy” as the first and “Finance OR Accounting OR Financial Performance OR Economic Performance OR Green Finance” as the second. Thus, the search was not restricted to accounting and finance. The phrase mentioned was corrected to clarify this point. Please, see the new explanation in Sections 3.1 and 3.2, and the details presented in Table 1. |
|
Suggestion 13: “Materials and Methods: Lines 244 - 245. Not understood. Implies that it excludes everything. Indicate the type of documents used and useful references to support your selection.” |
Dear reviewer, thank you for the observation. In the first filter application, by type of document, articles published in congresses and conferences, book chapters, books, and errata were excluded, but the documents type articles, review and editorial were kept, that it resulted in 246 documents. To clarify this point, we included Table 1 which presents search criteria in the Scopus Platform step by step. Please, see the additions in the manuscript. |
|
Suggestion 14: “Materials and Methods: Uses documents in English. Use references to validate what you have done.” |
Dear reviewer, thanks for the suggestion and observation. We agree that there are important papers written in English that could replace some references written in Portuguese that were used in our study. However, some of these papers were important during the development of this paper. Thus, although we added important references in English, as suggested, we opt to maintain some studies that were developed in Portuguese. |
|
Suggestion 15: “Materials and Methods: Why do you use all types of documents? Use references to validate what you have done.” |
Thank you for pointing out this topic. It is important to say that we used different types of documents to support our introduction, theoretical review, and some of our analysis, but the bibliometric and the systematic analyses, which were the core of the paper, was restricted to articles, review, and editorial published on the journals. The different types of documents were used as the aim to subside our argumentation, but as suggested, some of those references were changed. To meet your recommendation, we have restructured our whole Materials and Methods section.
|
|
Suggestion 16: “Materials and Methods: The use of bibliometric analyses is for the analysis of large data structures to study the intellectual structure of an academic discipline, field of study, journals, countries, etc. It is considered that at least 500 records should be used in this type of analysis; if less than 300 records are used, the quality of the information is not guaranteed (Donthu et al., 2021. 10.1016/j.jbusres.2021.04.070). The database used is less than 70 records (line 253). Therefore, the authors should consider conducting a literature review and not a bibliometric analysis. The study as conceived, would be misleading in nature and unreliable in the academic world.” |
Dear reviewer, thank you for pointing out this issue. To meet your recommendation, we have restructured our whole Materials and Methods section. We classified our research as a systematic literature review (SRL) and, in addition to the bibliometric analysis, we carried out a qualitative content analysis. The method followed the guidelines of important works about SRL such as of Thome et al. (2016), and Tranfield et al. (2003). Please, see the new section 3 in the manuscript and the new results added. |
|
Suggestion 17: “Materials and Methods: The data was downloaded in March 2021 (line 109). The year 2021 ended and presents incomplete information for that year. The latest trends could not be observed. The data needs to be updated.” |
Dear reviewer, thank you for the observation. As punctuated, the data was downloaded in March 2021, but the paper was developed during 2021 with the objective to be submitted to the special edition of the Sustainability journal that closed on December 2021. Thus, it was not possible the inclusion of all published papers from 2021. |
|
Suggestion 18: “Materials and Methods: Should indicate the usefulness of the software used. What are they for? .Moreover, which relevant studies have used the software.” |
Dear reviewer, thank you for the suggestion. We have further clarified our reasons to choose VosViewer software. We provided arguments based on previous studies and also examples of studies that applied the same software. Please, see the additions in the Section 3.3.
“The software used to assist data visualization and bibliometric analysis was the VOSviwer®, as it offers a detailed reading of bibliometric maps, in addition to supporting maps with more than 10 thousand items (van Eck and Waltman, 2009). Furthermore, Pan et al. (2018) found that VOSviewer is more frequently used and diffused than CiteSpace or HistCite in biblio-metric studies (e.g. Camón Luis and Celma, 2020; Cavalieri et al., 2021). Mendeley soft-ware was applied to read and organize the works analyzed, and Microsoft Excel to gener-ate graphs outside the database.” |
|
Suggestion 19: “Materials and Methods: The methodology does not include data cleaning. Can you indicate this?.” |
Dear Reviewer, thank you for the suggestion. We have restructured the Materials and Method within this point. Please see section 3. |
|
Suggestion 20: “Materials and Methods: When applying inclusion and exclusion criteria, you should report the number of documents that remain in the process.” |
Dear Reviewer, thank you for the suggestion. We made a table with the conducted steps and filtering criteria. Please see Table 1 in Section 3.2. |
|
Suggestion 21: “Results: Most of the analyses are purely descriptive, making the paper of no academic value. Presenting figures or tables and describing them is not research work. Analyses should guide the reader to more detailed information about the manuscripts, authors, countries, and topics. Authors should provide an analysis of the content of each figure and indicate the significance of the data presented. I recommend exploring papers that do this type of bibliometric analysis. I attach an example of the hundreds of papers available online; these are MDPI and are recent according to the advances in bibliometrics (Giraldo et al. 2019 10.3390/agronomy9070352 or Carrion et al. 2021 10.3390/ijerph18189445). It should set out the contribution of each analysis to the academy. These articles set out a very appropriate structure for a Bibliometric analysis. They deal in a very interesting way with methodology, results, and discussion. The authors present several studies in Bibliometrics that can serve as examples.” |
Dear Reviewer, thank you for the suggestion. We reorganized the text and to improve the analysis we include more details about the systematic review. Please see section 4, Table 6, and appendix. |
|
Suggestion 22: “Results: Table 1 shows errors. For example, if it presents 69 documents (line 253), the table exceeds 150 documents.” |
Dear Reviewer, thank you for pointing it out. We corrected it with the right figures. Please see Table 2. |
|
Suggestion 23: “Results: Table 2 does not show relevant information from journals such as Citescore, SJR, and H-Index indicators.” |
Dear Reviewer, thank you for your suggestion. We inserted in Table 3 (old Table 2) SJR and H-Index, and we did a brief explanation about these indicators, as suggested. |
|
Suggestion 24: “Results: Table 3 is poor, as is the analysis. For example, the analysis should be interesting: What kind of research does China carry out with the three main collaborating countries? Why are most of the production centred in Europe? Etc.” |
Dear Reviewer, thank you for your suggestion. We have punctuated the importance of China and Europe for the subject in section 4.3 as suggested. |
|
Suggestion 25: “Results: I think it is sufficient to give examples. Authors should improve their analysis and content.” |
Dear Reviewer, thank you for all your suggestions. To meet your recommendation, we have restructured our whole Materials and Methods section, and we improved the performance analysis. Please see Sections 3, 4, Table 6, and appendix. |
|
Suggestion 26: “Discussion and conclusions: The discussion section lives up to its name because the authors have to discuss the results found in the research. Here, should cross-reference the figures and tables presented as generalisations of the study with the theory to determine whether aspects can affirm the existing theory or present new findings.” |
Dear Reviewer 2, thank you for the important suggestions and for your contribution to improving the quality of our study. We restructured the section "Introduction", "Materials and Methods", and "analyses" with the objective of improving the paper, showing the new findings, and attending to your suggestions. We hope to have addressed all of your comments satisfactorily. |
|
Suggestion 27: “Discussion and conclusions: Conclusions should be brief and highlight important findings, limitations or future lines of research. References should not be displayed unless it is an extraordinary case.” |
Dear Reviewer 2, thank you for your observation and for all your suggestions. |

Reviewer 2 Report
The manuscript suggests a current and attractive topic for academia. The effort made is evident, but it presents conceptual errors in the methodology, an inadequate search for terms and poor analysis, which do not allow it to meet the standards of bibliometric studies. I attach some observations that I hope will be useful in improving the manuscript:
Abstract
1) This section needs to be rewritten. The abstract should contain concisely in a single paragraph the importance of the study, the objective, methodology, results, discussion and conclusions.
Introduction
2) In general terms, should revise the introduction. The literature on Circular Economy and Financial Performance is very poor. It shows few references and a lot of unsubstantiated text, which is incompatible for a prestigious journal. Moreover, the few references used do not show authors of reference in the study area.
3) It does not state the study's objective (it should indicate the word aim).
4) It should indicate the importance of the study for the academy. In other words, a bibliometric study.
5) Do not mention relevant studies in the area ("circular economy", "bibliometric"). For example Worldwide research on circular economy and environment: A bibliometric analysis (Ruiz-Real et al. 2018), research insights (Goyal et al., 2021), Circular Economy. A review and bibliometric analysis (Camón & Celma), The circular economy umbrella: Trends and gaps on integrating pathways ( Homrich & Galvao, 2018). Or together ("Circular economy & performance") Nexus of circular economy and sustainable business performance in the era of digitalization (Agrawal et al., 2021), Circular economy performance assessment methods: A systematic literature review (Sassanelli et al., 2019). Regarding the last two references, what is special about the study to make it relevant when similar studies exist?
6) The authors need to state how the article is going to be divided (number of sections) and what each section comprises.
Theoretical Foundation
7) There is an interesting section on the circular economy (from a historical point of view) rather than on business application (the latter should be the central point due to the subject matter). The curious thing is that "financial performance" (the second important term), which is mentioned in the title, does not appear in this section or is not mentioned at all. Therefore, the authors should agree on the content of the body or the title because it is not related.
Materials and Methods
This section requires complete restructuring, expansion and new references (there are practically no references for a modern and fashionable topic). Bibliometric studies have a fundamental characteristic; they must be systematic, transparent and reproducible. As far as possible, a sequence of methodological stages (steps) and a figure summarising the process should be indicated.
8) Lines 177 - 178. The above definition is almost the classic definition of Alan Pritchard from the 1960s. Use more renowned references.
9) Line 176 indicates two divisions. This is conceptually flawed. Bibliometric analyses are divided into performance and mapping. See Cobo et al. 2011 "An approach for detecting, quantifying, and visualizing the evolution of a research field: A practical application to the Fuzzy Sets Theory field".
10) Why use Scopus? There are other important databases such as Web of science and Dimensions. Use arguments and references to defend your idea. Do not use the argument of Scopus itself; it must be of scientific articles.
11) Lines 230 - 232. The search is questionable. The title is about "financial performance" but is not used here. It adds terms "Finance, "Accounting", which are not necessarily related to performance. This search is questionable and unreliable.
12) Lines 244 - 245. Not understood. Implies that it excludes everything. Indicate the type of documents used and useful references to support your selection.
13) Uses documents in English. Use references to validate what you have done.
14) Why do you use all types of documents? Use references to validate what you have done.
15) The use of bibliometric analyses is for the analysis of large data structures to study the intellectual structure of an academic discipline, field of study, journals, countries, etc. It is considered that at least 500 records should be used in this type of analysis; if less than 300 records are used, the quality of the information is not guaranteed (Donthu et al., 2021. 10.1016/j.jbusres.2021.04.070). The database used is less than 70 records (line 253). Therefore, the authors should consider conducting a literature review and not a bibliometric analysis. The study as conceived, would be misleading in nature and unreliable in the academic world.
16) The data was downloaded in March 2021 (line 109). The year 2021 ended and presents incomplete information for that year. The latest trends could not be observed. The data needs to be updated.
17) Should indicate the usefulness of the software used. What are they for? .Moreover, which relevant studies have used the software.
18) The methodology does not include data cleaning. Can you indicate this?
19) When applying inclusion and exclusion criteria, you should report the number of documents that remain in the process.
Results
20) Most of the analyses are purely descriptive, making the paper of no academic value. Presenting figures or tables and describing them is not research work. Analyses should guide the reader to more detailed information about the manuscripts, authors, countries, and topics. Authors should provide an analysis of the content of each figure and indicate the significance of the data presented. I recommend exploring papers that do this type of bibliometric analysis. I attach an example of the hundreds of papers available online; these are MDPI and are recent according to the advances in bibliometrics (Giraldo et al. 2019 10.3390/agronomy9070352 or Carrion et al. 2021 10.3390/ijerph18189445). It should set out the contribution of each analysis to the academy. These articles set out a very appropriate structure for a Bibliometric analysis. They deal in a very interesting way with methodology, results, and discussion. The authors present several studies in Bibliometrics that can serve as examples.
Below are some examples:
21) Table 1 shows errors. For example, if it presents 69 documents (line 253), the table exceeds 150 documents.
22) Table 2 does not show relevant information from journals such as Citescore, SJR, and H-Index indicators.
23) Table 3 is poor, as is the analysis. For example, the analysis should be interesting: What kind of research does China carry out with the three main collaborating countries? Why are most of the production centred in Europe? Etc.
24) I think it is sufficient to give examples. Authors should improve their analysis and content.
Discussion and conclusions
25) The discussion section lives up to its name because the authors have to discuss the results found in the research. Here, should cross-reference the figures and tables presented as generalisations of the study with the theory to determine whether aspects can affirm the existing theory or present new findings.
26) Conclusions should be brief and highlight important findings, limitations or future lines of research. References should not be displayed unless it is an extraordinary case.
Author Response

(The authors gave the same response as above.)

Round 2
Reviewer 2 Report
The manuscript suggests a current and attractive topic for academia. It is evident that efforts have been made to improve the paper and that some of this reviewer's suggestions have been partially realised.
The main problem with the article is its conceptual nature. In offering a systematic literature review, the authors should exhibit the characteristic elements of this type of study. Unfortunately, they present a manuscript with a mixture of scholarly reporting, bibliometrics and almost nothing of a systematic literature review.
I emphasise what was stated in the first review:
"The use of bibliometric analyses is for the analysis of large data structures to study the intellectual structure of an academic discipline, field of study, journals, countries, etc. It is considered that at least 500 records should be used in this type of analysis; if less than 300 records are used, the quality of the information is not guaranteed (Donthu et al., 2021. 10.1016/j.jbusres.2021.04.070). The database used is less than 70 records (line 253). Therefore, the authors should consider conducting a literature review and not a bibliometric analysis. The study as conceived, would be misleading in nature and unreliable in the academic world".
In other words, if you present maps constructed with less than 300 records, the information is not reliable (especially author keywords). It is not a question of changing the title; you must change the document's content. If you offer a systematic review, this should be observed. It is possible to combine both as long as the emphasis is on literature review and the bibliometric maps used are supportive (such as countries and authors involved). Examples of this are attached: Bartolini et al. 2019. 10.1016/j.jclepro.2019.04.055; Fahimia & Sarkis 2015. 10.1016/j.ijpe.2015.01.003.
An article is not attaching informative tables (Table 6 and Appendix A) and thereby claiming that it is a literature review. These types of tables are made when the researcher explores a topic as part of their portfolio and not to be inserted into an article.
I recommend that you review these comments and the first review. Then, authors should agree and submit a manuscript according to its conceptual nature (literature review, bibliometric or a combination, although the latter is a very complex work).
Good luck.
Author Response
Manuscript ID sustainability-1557257
Special Issue: Sustainable Finance and Value Creation
“Circular Economy and Financial Aspects: A Systematic Review of the Literature”
Dear Editor,
We are thankful to you for giving us another chance to improve our paper. Please, see the enclosed itemized reply to the reviewer. The additions are highlighted in the manuscript using Track Changes/Comments tools of the Word software. We are sending two versions of our manuscript: a tracked-change version and a clean version.
With Kind Regards,
The Authors
February 16, 2022
Response to Reviewers’ Comments
Reviewer #2
General comment: “The manuscript suggests a current and attractive topic for academia. It is evident that efforts have been made to improve the paper and that some of this reviewer's suggestions have been partially realised.”
Reply: Dear Reviewer, thank you for recognizing our effort in improving the paper. We hope to have fulfilled your suggestions at this round of revision.
Suggestion 1: “ The use of bibliometric analyses is for the analysis of large data structures to study the intellectual structure of an academic discipline, field of study, journals, countries, etc. It is considered that at least 500 records should be used in this type of analysis; if less than 300 records are used, the quality of the information is not guaranteed (Donthu et al., 2021. 10.1016/j.jbusres.2021.04.070). The database used is less than 70 records (line 253). Therefore, the authors should consider conducting a literature review and not a bibliometric analysis. The study as conceived, would be misleading in nature and unreliable in the academic world".
In other words, if you present maps constructed with less than 300 records, the information is not reliable (especially author keywords). It is not a question of changing the title; you must change the document's content. If you offer a systematic review, this should be observed. It is possible to combine both as long as the emphasis is on literature review and the bibliometric maps used are supportive (such as countries and authors involved). Examples of this are attached: Bartolini et al. 2019. 10.1016/j.jclepro.2019.04.055; Fahimia & Sarkis 2015. 10.1016/j.ijpe.2015.01.003.”
Reply: Dear Reviewer,
Thank you very much for your comments that significantly improved our paper. In this study, we conducted a systematic literature review, using both quantitative and qualitative methods of analysis. While for the quantitative approach, we applied a bibliometric analysis to map the field and the relationships between research constituents (Donthu et al., 2021), in the qualitative one, we used the content analysis to gain insights and to deepen the understanding in the topic. As pointed out by Donthu et al. (2021), “systematic literature reviews are better suited for confined or niche research areas”, which is the case of our study focusing on the specific topic of finance and the circular economy.
Donthu et al. (2021) also explain that “scholars should take extra care when making qualitative assertions about bibliometric observations and supplement them with content analysis, where appropriate (Gaur & Kumar, 2018)”. In this sense, our study applies bibliometric and content analyses as complementary methods to achieve our aim of analyzing the existing studies that investigate the link between the circular economy (CE) and financial aspects. We used the network analysis to map the scientific area that studies the CE and financial aspects, and then the content analysis to synthesize and gain knowledge on the field.
At this point, we would like to clarify that we understand that using only the bibliometric analysis for a small sample could be considered overkill (Donthu et al., 2021), and this method is more suitable when it is applied as a unique method in the study for analyzing a large sample of articles. That is, when the methodology is only based on bibliometrics and quantitative approach, then the sample has to be large to bring meaningful discussions. Nevertheless, in our case, we aimed to make qualitative assertions. That is the reason why we have discussed the networks generated by analyzing the content of the articles present in the network and by performing a content analysis in the whole sample of 69 papers.
The approach we took is not uncommon in the literature. Thomé et al. (2016) that explain how to conduct a systematic literature review (SLR), reinforce that a qualitative SLR is often combined with quantitative methods, such as the bibliometric analysis to identify and visualize contemporary and emerging themes. To support this argument and show that, when using a combination of bibliometric and content analysis in a SLR, it is not necessary a sample larger than 300 records, we present examples of some existing works.
(1) Jugend et al. (2020) mixed bibliometric analysis and content analysis to study public support for innovation with a dataset of 121 articles – doi 10.1016/j.techfore.2020.119985 ;
(2) Thomas and Gupta (2021) conducted a bibliometric and a framework-based systematic review, refining the sample by applying inclusion and exclusion criteria from 1.449 to 462 articles, then 142, and finally achieving the final sample of 59 studies that were meaningful for their topic, which was used to conduct the analysis – doi 10.1108/JKM-01-2021-0026 ;
(3) Ferreira et al. (2018) applied a bibliometric analysis and systematic review of the literature using a dataset of 53 articles, after successive refinements in the initial sample - doi 10.1504/IJLSM.2018.091445.
Regarding the comment “In other words, if you present maps constructed with less than 300 records, the information is not reliable (especially author keywords)”, we would like to clarify that it might be true in cases of making assumptions for a broad area such as only the “Circular Economy”, because small samples would not represent the entire diversity of topics which is encompassed by the CE. However, in the case of narrowing the focus to a specific scope such as CE and financial aspects, the statement does not hold. It is possible to apply the bibliometric method for small samples and bring meaningful discussions for that emerging field. For example, Ferreira et al. (2021) studied sustainability in family business (a narrow scope) using only bibliometric analysis in a sample of 161 articles – doi 10.1016/j.techfore.2021.121077 . Besides, when you mention “especially author keywords”, we need to explain that the co-word analysis is a technique that examines the actual content of the publication itself (Donthu et al., 2021). Therefore, it generates a reliable analysis based on the content of the sample, being small or not. That is, the network represents the content of articles of the sample, which is the focus of the study. We agree that we could not generalize our findings and discussion to a great extent, but we do could make assumptions based on the scope of our sample.
Overall, what we want to show you is that the methodology applied in our study, combining the bibliometric and content analyses for a systematic literature review is reliable, reasonable, and it is common in the scientific literature. To make our point clear in the manuscript, we made additions in the Method section and also expanded our discussions based on the content analysis to improve the review and support our conclusions.
We appreciate your valuable time in assessing the paper, and we hope to have clarified our point of view.
Suggestions 2 and 3: “The main problem with the article is its conceptual nature. In offering a systematic literature review, the authors should exhibit the characteristic elements of this type of study. Unfortunately, they present a manuscript with a mixture of scholarly reporting, bibliometrics and almost nothing of a systematic literature review.
An article is not attaching informative tables (Table 6 and Appendix A) and thereby claiming that it is a literature review. These types of tables are made when the researcher explores a topic as part of their portfolio and not to be inserted into an article.””
Reply: Dear Reviewer,
Thanks for your comments. We would like to clarify that our article is denominated as a literature review since it consolidates the results of different studies on the subject of “Circular Economy and Financial Aspects” with the aim of achieving greater understanding and reaching a level of conceptual or theoretical development (Campbell et al., 2003). Our review is classified as systematic (SRL) because it adopts a replicable, scientific and transparent process, by providing a trail of decisions and procedures taken by the researchers. Each step and procedure taken are detailed in the Methodology section. Besides, it offers relevant contributions and discussions by synthesizing the findings of the articles in our sample, which is a characteristic of literature reviews. According to Tranfield et al. (2005), a literature review presents two main forms of analyzing and interpreting the results. First, in a descriptive manner, detailed through data extraction categories and codes. Second, as a thematic analysis, discussing results derived or not from an aggregative or interpretive approach, outlining what is known and established from the contributions of the studies (similarities and differences). In that way, we strongly believe that our study was capable of achieving the contributions expected from a SLR through the discussions and conclusions presented in Sections 4 and 5, and also by further expanding the analysis of the Table classifying the 69 articles of our sample. Overall, we were able to show what is already known about the circular economy and financial aspects by discussing the contributions of each study analyzed, and also to present what is not known about the topic yet.
In order to attend to the suggestions, table 6 was excluded from the text and its contents were summarized in section 4.7. This section resumes the systematization and contributions of the analyzed studies. Please, see the additions in the manuscript.
References
Campbell, R., et al. Evaluating meta-ethnography: a synthesis of qualitative research on lay experiences of diabetes and diabetes care. Social Science & Medicine, 2003, 53, 671-684, doi:10.1016/S0277-9536(02)00064-3.
Donthu, N. et al. How to conduct a bibliometric analysis: An overview and guidelines. Journal of Business Research, 2021, 133, 285-296, doi:10.1016/j.jbusres.2021.04.070.
Ferreira, K. A.; Flávio, L. A.; Rodrigues, L. F. Postponement: bibliometric analysis and systematic review of the literature. International Journal of Logistics Systems and Management, 2018, 30, 69-94, doi: 10.1504/IJLSM.2018.091445.
Jugend, D. et al. Public support for innovation: A systematic review of the literature and implications for open innovation. Technological Forecasting and Social Change, 2020, 156, 119985, doi:10.1016/j.techfore.2020.119985.
Thomas, A.; Gupta, V. Tacit knowledge in organizations: bibliometrics and a framework-based systematic review of antecedents, outcomes, theories, methods and future directions. Journal of Knowledge Management, 2021, doi:10.1108/JKM-01-2021-0026.

Round 3
Reviewer 2 Report
This is an interesting manuscript that suggests a current and attractive topic for the academy. The effort made by presenting an interesting document that has undergone some modifications due to the evaluators' recommendations has allowed us to have a novel and rich document on a complex subject.
I appreciate the authors' patience when considering the vast majority of the recommendations made with quality and professionalism. The methodology and the data set as its analyzes are solid. The conclusions are relevant.
My sincere congratulations to the authors for this important contribution to academia. I consider the article to be publishable.